# PD-MORL: Preference-Driven Multi-Objective Reinforcement Learning Algorithm

**Toygun Basaklar**
UW-Madison
Madison, WI 53706
basaklar@wisc.edu

**Suat Gumussoy**
Siemens Corporate Technology
Princeton, NJ 08540
suat.gumussoy@siemens.com

**Umit Y. Ogras**
UW-Madison
Madison, WI 53706
uogras@wisc.edu

## Abstract

Multi-objective reinforcement learning (MORL) approaches have emerged to tackle many real-world problems with multiple conflicting objectives by maximizing a joint objective function weighted by a preference vector. These approaches find *fixed customized policies corresponding to preference vectors specified during training*. However, the design constraints and objectives typically change dynamically in real-life scenarios. Furthermore, storing a policy for each potential preference is not scalable. Hence, obtaining a set of Pareto front solutions for the entire preference space in a given domain with a single training is critical. To this end, we propose a novel MORL algorithm that trains a *single universal network to cover the entire preference space scalable to continuous robotic tasks*. The proposed approach, Preference-Driven MORL (PD-MORL), utilizes the preferences as guidance to update the network parameters. It also employs a novel parallelization approach to increase sample efficiency. We show that PD-MORL achieves up to 25% larger hypervolume for challenging continuous control tasks and uses an order of magnitude fewer trainable parameters compared to prior approaches.

## 1 Introduction

Reinforcement learning (RL) has emerged as a promising approach to solve various challenging problems including board/video games (Silver et al., 2016; Mnih et al., 2015; Shao et al., 2019), robotics (Nguyen & La, 2019), smart systems (Gupta et al., 2020; Yu et al., 2021), and chip design/placement (Zheng & Louri, 2019; Mirhoseini et al., 2020). The main objective in a standard RL setting is to obtain a policy that maximizes a single cumulative reward by interacting with the environment. However, many real-world problems involve multiple, possibly conflicting, objectives. For example, robotics tasks should maximize speed while minimizing energy consumption. In contrast to single-objective environments, performance is measured using multiple objectives. Consequently, there are multiple Pareto-optimal solutions as a function of the preference between objectives (Navon et al., 2020). Multi-objective reinforcement learning (MORL) approaches (Hayes et al., 2022) have emerged to tackle these problems by maximizing a vector of rewards depending on the preferences.

Existing approaches for multi-objective optimization generally transform the multidimensional objective space into a single dimension by statically assigning weights (preferences) to each objective (Liu et al., 2014). Then, they use standard RL algorithms to obtain a policy optimized for the given preferences. These approaches suffer when the objectives have widely varying magnitudes since setting the preference weights requires application domain expertise. More importantly, they can find only a single solution for a given set of goals and constraints. Thus, they need to repeat the training progress when the constraints or goals change. However, repetitive retraining is impractical since the constraints and design can change frequently depending on the application domain. Therefore, obtaining a set of Pareto front solutions that covers the entire preference space with a single training is critical (Xu et al., 2020; Yang et al., 2019; Abdolmaleki et al., 2020).

This paper presents a novel multi-objective reinforcement learning algorithm *using a single policy network that covers the entire preference space scalable to continuous robotic tasks*. At its core, it uses a multi-objective version of Q-Learning, where we approximate the Q-values with a neural network. This network takes the states and preferences as inputs during training. Making the preferences input parameters allows the trained model to produce the optimal policy for any user-specified preference at run-time. Since the user-specified preferences effectively drive the policy decisions, it is called

preference-driven (PD) MORL. For each episode during training, we randomly sample a preference vector ($\boldsymbol{\omega} \in \Omega : \sum_{i=0}^{L} \omega_i = 1$) from a uniform distribution. Since the number of collected transitions by interacting with the environment for some preferences may be underrepresented, we utilize hindsight experience replay buffer (HER) (Andrychowicz et al., 2017). As a key insight, we observe that the preference vectors have similar directional angles to the corresponding vectorized Q-values for a given state. Using the insight, we utilize the cosine similarity between the preference vector and vectorized Q-values in the Bellman's optimality operator to guide the training. However, not every Pareto front perfectly aligned with the preference vectors. To mitigate this adverse effect, we fit a multi-dimensional interpolator to project the original preference vectors ($\boldsymbol{\omega} \in \Omega$) to normalized solution space to align preferences with the multi-objective solutions. The projected preference vectors are used in our novel preference-driven optimality operator to obtain the target Q-values. Additionally, to increase the sample efficiency of the algorithm, we divide the preference space into sub-spaces and assign a child process to these sub-spaces. Each child process is responsible for its own preference sub-space to collect transitions. This parallelization provides efficient exploration during training, assuring that there is no bias towards any preference sub-space.

PD-MORL can be employed in any off-policy RL algorithm. We develop a multi-objective version of the double deep Q-network algorithm with hindsight experience replay buffer (MO-DDQN-HER) (Schaul et al., 2015) for problems with discrete action spaces and evaluate PD-MORL's performance on two commonly used MORL benchmarks: Deep Sea Treasure (Hayes et al., 2022) and Fruit Tree Navigation Task (Yang et al., 2019). We specifically choose these two benchmarks to make a fair comparison with the prior approach (Yang et al., 2019) that also aims to achieve a unified policy network. Additionally, we develop a multi-objective version of the Twin Delayed Deep Deterministic policy gradient algorithm with hindsight experience replay buffer (MO-TD3-HER) (Fujimoto et al., 2018a) for problems with continuous action spaces. Using MO-TD3-HER, we evaluate PD-MORL on the multi-objective continuous control tasks such as MO-Walker2d-v2, MO-HalfCheetah-v2, MO-Ant-v2, MO-Swimmer-v2, MO-Hopper-v2 that are presented by Xu et al. (2020). With the combination of the use of the cosine similarity term, the HER, and the parallel exploration, PD-MORL achieves up to 78% larger hypervolume for simple benchmarks and 25% larger hypervolume for continuous control tasks and uses an order of magnitude fewer trainable parameters compared to prior approaches while achieving broad and dense Pareto front solutions. We emphasize that it achieves these results with a single policy network.

## 2 RELATED WORK

Existing MORL approaches can be classified as single-policy, multi-policy, and meta-policy approaches. The main difference among them is the number of policies learned during training. Single-policy approaches transform a multi-objective problem into a single-objective problem by combining the rewards into a single scalar reward using a scalarization function. Then, they use standard RL approaches to maximize the scalar reward (Roijers et al., 2013). Most of the previous studies find the optimal policy for *a given preference* between the objectives using scalarization (Van Moffaert et al., 2013). Additionally, recent work takes an orthogonal approach and encodes preferences as constraints instead of scalarization (Abdolmaleki et al., 2020). These approaches have two primary drawbacks: they require domain-specific knowledge, and preferences must be set beforehand.

Multi-policy approaches aim to obtain a set of policies that approximates the Pareto front of optimal solutions. The most widely used approach is to repeatedly perform a single-policy algorithm over various preferences (Roijers et al., 2014; Mossalam et al., 2016; Zuluaga et al., 2016). However, this approach suffers from a large number of objectives and dense Pareto solutions for complex control problems. In contrast, Pirotta et al. (2015) suggest a manifold-based policy search MORL approach which assumes policies to be sampled from a manifold. This manifold is defined as a parametric distribution over the policy parameter space. Their approach updates the manifold according to an indicator function such that sample policies yield an improved Pareto front. Parisi et al. (2017) extend this method with hypervolume and non-dominance count indicator functions using importance sampling to increase the sample efficiency of the algorithm. However, the number of parameters to model the manifold grows quadratically (Chen et al., 2019) as the number of policy parameters increases. Therefore, these approaches are not scalable for complex problems such as continuous control robotics tasks to achieve a dense Pareto front where deep neural networks with at least thousands of parameters are needed.

Abels et al. (2019) and Yang et al. (2019) use a multi-objective Q-learning approach that simultaneously learns a set of policies over multiple preferences. These studies use a single network that takes preferences as inputs and uses vectorized value function updates in contrast to standard scalarized value function updates. An orthogonal approach by Chen et al. (2019) is the meta-policy approach that frames MORL as a meta-learning problem using a task distribution given by a distribution over the preference space. The authors first train a meta-policy to approximate the Pareto front implicitly. Then, they obtain the Pareto optimal solution of a given preference by only fine-tuning the meta-policy with a few gradient updates. A more recent study proposes an efficient evolutionary learning algorithm to update a population of policies simultaneously in each run to improve the approximation of the Pareto optimal solutions (Xu et al., 2020). All non-dominated policies for each generation of policies are stored as an external Pareto front archive during the training. Finally, it outputs this Pareto front archive as approximated Pareto front solutions. However, this approach obtains a policy network for each solution on the Pareto front. Moreover, there is no specific knowledge of correspondence between preference vectors and solutions in this Pareto front.

Distinct from the prior work, PD-MORL utilizes the relation between the Q-values and the preferences and proposes a novel update rule, including the cosine similarity between the Q-values and preferences. Additionally, PD-MORL increases its sample efficiency by using a novel parallelization approach with HER and provides efficient exploration. The combination of the cosine similarity term, the HER, and the parallel exploration achieve up to 25% larger hypervolume for challenging continuous control tasks compared to prior approaches using an order of magnitude fewer trainable parameters.

## 3 BACKGROUND

MORL requires learning several tasks with different rewards simultaneously. Each objective has an associated reward signal, transforming the reward from a scalar to a vector, $\mathbf{r} = [r_1, r_2, ..., r_L]^T$, where $L$ is the number of objectives. A multi-objective problem can be formulated as a multi-objective Markov decision process (MOMDP) defined by the tuple $\langle \mathcal{S}, \mathcal{A}, \mathcal{P}, \mathbf{r}, \Omega, f_\omega \rangle$, where $\mathcal{S}, \mathcal{A}, \mathcal{P}(s'|s, a)$, $\mathbf{r}$, and $\Omega$ represents state space, action space, transition distribution, reward vector, and preference space, respectively. The function $f_\omega(\mathbf{r}) = \omega^T \mathbf{r}$ yields a scalarized reward using a preference of $\omega \in \Omega$. If $\omega$ is taken as a fixed vector, the MOMDP boils down to a standard MDP, which can be solved using standard RL techniques. However, if we consider all possible returns from a MOMDP and all possible preferences in $\Omega$, a set of non-dominated policies called the Pareto front can be obtained. A policy $\pi$ is Pareto optimal if there is no other policy $\pi'$ that improves its expected return for an objective without degrading the expected return of any other objective. For complex problems, such as continuous control tasks, obtaining an optimal Pareto front using an RL setting is an NP-hard problem (Xu et al., 2020). Hence, the main goal of the MORL algorithms is to obtain an approximation of the Pareto front. The quality of the approximated Pareto front is typically measured by two metrics: (i) hypervolume and (ii) sparsity (Hayes et al., 2022).

**Definition 1** (Hypervolume Indicator). Let $P$ be a Pareto front approximation in an L-dimensional objective space and contains $N$ solutions. Let $\mathbf{r}_0 \in \mathbb{R}^L$ be the reference point. Then, the hypervolume indicator is defined as:

$$I_H(P) := \Lambda(H(P, \mathbf{r}_0)) \qquad (1)$$

where $H(P, \mathbf{r}_0) = \{\mathbf{z} \in \mathbb{R}^L | \exists\ 1 \leq i \leq N : \mathbf{r}_0 \preceq \mathbf{z} \preceq P_i\}$ with $P_i$ being the $i^{th}$ solution in $P$ and $\preceq$ is the relation operator of multi-objective dominance. $\Lambda(\cdot)$ denotes the Lebesgue measure with $\Lambda(H(P, \mathbf{r}_0)) = \int_{\mathbb{R}^L} \mathbb{1}_{H(P, \mathbf{r}_0)}(\mathbf{z}) d\mathbf{z}$ and $\mathbb{1}_{H(P, \mathbf{r}_0)}$ being the characteristic function of $H(P, \mathbf{r}_0)$.

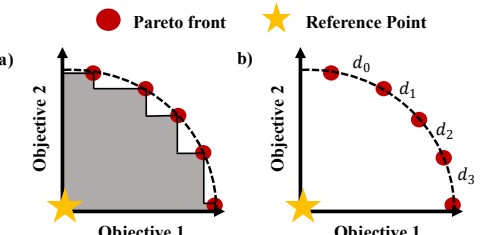

Figure 1: (a) The hypervolume between a reference point and solutions in Pareto front is shown with the shaded area. (b) The sparsity is the average square distance ($d_i, i = \{0, \ldots, 3\}$) between consecutive solutions in the Pareto front.

**Definition 2** (Sparsity). For the same $P$, sparsity is defined as:

$$Sp(P) := \frac{1}{N-1} \sum_{j=1}^{L} \sum_{i=1}^{N-1} (P_{i_j} - P_{i+1_j})^2 \qquad (2)$$

where $P_i$ is the $i^{th}$ solution in $P$ and $P_{i_j}$ is the sorted set for the $j^{th}$ objective.

Improvement in any objective manifests as an increase in the hypervolume, as illustrated in Figure 1. However, the hypervolume alone is insufficient to assess the quality of the Pareto front. For instance, hypervolume may increase due to an increase in only one of the objectives which indicates that the algorithm does not improve other objectives. The sparsity metric is introduced to assess whether the Pareto front solutions are dense or sparse. A larger hypervolume indicates that a more desired Pareto front approximation is achieved. Lower sparsity values imply that a dense set of solutions is achieved.

## 4 PD-MORL: PREFERENCE DRIVEN MORL ALGORITHM

This section introduces the proposed preference-driven MORL algorithm. We first provide a theoretical analysis of PD-MORL based on the multi-objective version of Q-learning (Watkins & Dayan, 1992) by following the framework provided by Yang et al. (2019). The proofs for this theoretical analysis are available in Appendix A. We then introduce multi-objective double deep Q-networks with a preference-driven optimality operator for problems with discrete action space. Finally, we extend the Twin Delayed Deep Deterministic Policy Gradient (TD3) algorithm (Fujimoto et al., 2018b) to a multi-objective version utilizing our proposed approach for problems with continuous action space.

### 4.1 THEORETICAL ANALYSIS OF PD-MORL

In standard Q-learning, the value space is defined as $\mathcal{Q} \in \mathbb{R}^{\mathcal{S} \times \mathcal{A}}$, containing all bounded functions $Q(s, a)$ which are the estimates of the total expected rewards when the agent is at state $s$, taking action $a$. We extend it to a multi-objective value space by defining the value space as $\mathcal{Q} \in \mathbb{R}^{L^{\mathcal{S} \times \mathcal{A}}}$, containing all bounded functions $\mathbf{Q}(s, a, \boldsymbol{\omega})$ which are the estimates of expected total rewards under preference $\boldsymbol{\omega} \in \mathbb{R}^L_{\geq 0} : \sum_{i=0}^L \omega_i = 1$. We then define a metric in this value space as:

$$d(\mathbf{Q}, \mathbf{Q}') := \sup_{s \in \mathcal{S}, a \in \mathcal{A}, \boldsymbol{\omega} \in \Omega} |\boldsymbol{\omega}^T(\mathbf{Q}(s, a, \boldsymbol{\omega}) - \mathbf{Q}'(s, a, \boldsymbol{\omega}))|. \tag{3}$$

This metric gives the distance between $\mathbf{Q}$ and $\mathbf{Q}'$ as the supremum norm of the scalarized distance between these two vectors where the identity of indiscernables ($d(\mathbf{Q}, \mathbf{Q}') = 0 \Leftrightarrow \mathbf{Q} = \mathbf{Q}'$) does not hold and thus, makes metric $d$ a pseudo-metric. Further details for the axioms of this pseudo-metric are given in Appendix A. In the following Theorem, we show that the metric space equipped with this metric is complete since the limit of the sequence of operators is required to be in this space.

**Theorem 1** (Multi-objective Metric Space $(\mathcal{Q}, d)$ is Complete). *The metric space $(\mathcal{Q}, d)$ is complete and every Cauchy sequence $\mathbf{Q}_n(s, a, \boldsymbol{\omega})$ is convergent in metric space $(\mathcal{Q}, d) \, \forall \, s, a, \boldsymbol{\omega} \in \mathcal{S}, \mathcal{A}, \Omega$.*

Given a policy $\pi$ and sampled transition $\tau$, we can define multi-objective Bellman's evaluation operator $\mathcal{T}_\pi$ using the metric space $(\mathcal{Q}, d)$ as:

$$(\mathcal{T}_\pi \mathbf{Q})(s, a, \boldsymbol{\omega}) := \mathbf{r}(s, a) + \gamma \mathbb{E}_{\mathcal{T} \sim (\mathcal{P}, \pi)} \mathbf{Q}(s', a', \boldsymbol{\omega}) \tag{4}$$

where $(s', a', \boldsymbol{\omega})$ denotes the next state-action-preference pair and $\gamma \in (0, 1)$ is the discount factor.

We define a *preference-driven optimality operator* $\mathcal{T}$ by adding a cosine similarity term between preference vectors and state-action values to the multi-objective Bellman's optimality operator as:

$$(\mathcal{T}\mathbf{Q})(s, a, \boldsymbol{\omega}) := \mathbf{r}(s, a) + \gamma \mathbb{E}_{s' \sim \mathcal{P}(\cdot|s, a)} \mathbf{Q}(s', \sup_{a' \in \mathcal{A}} (S_c(\boldsymbol{\omega}, \mathbf{Q}(s', a', \boldsymbol{\omega})) \cdot (\boldsymbol{\omega}^T \mathbf{Q}(s', a', \boldsymbol{\omega}))), \boldsymbol{\omega}) \tag{5}$$

where $S_c(\boldsymbol{\omega}, \mathbf{Q}(s', a', \boldsymbol{\omega}))$ denotes the cosine similarity between preference vector and Q-value. This term enables our optimality operator to choose actions that align the preferences with the Q-values and maximize the target value, as elaborated in Section 4.2 under the preference alignment subtitle.

**Theorem 2** (Multi-objective Bellman's Evaluation Operator is Contraction). *Let $(\mathcal{Q}, d)$ be a complete metric space (as in Theorem 1). Let $\mathbf{Q}$ and $\mathbf{Q}'$ be any two multi-objective Q-value functions in this space. The multi-objective Bellman's evaluation operator is a contraction and $d(\mathcal{T}_\pi \mathbf{Q}, \mathcal{T}_\pi \mathbf{Q}') \leq \gamma d(\mathbf{Q}, \mathbf{Q}')$ holds for the Lipschitz constant $\gamma$ (the discount factor).*

**Theorem 3** (Preference-driven Multi-objective Bellman's Optimality Operator is Contraction). *Let $(\mathcal{Q}, d)$ be a complete metric space. Let $\mathbf{Q}$ and $\mathbf{Q}'$ be any two multi-objective Q-value functions in this space. The preference-driven multi-objective Bellman's optimality operator is a contraction and $d(\mathcal{T}\mathbf{Q}, \mathcal{T}\mathbf{Q}') \leq \gamma d(\mathbf{Q}, \mathbf{Q}')$ holds for the Lipschitz constant $\gamma$ (the discount factor).*

Theorem 2 and Theorem 3 state that multi-objective evaluation and optimality operators are contractions. They ensure that we can apply our optimality operator in Equation 5 iteratively to obtain the optimal multi-objective value function given by Theorem 4 and Theorem 5 below, respectively.

**Theorem 4** (Preference-driven Multi-objective Optimality Operator Converges to a Fixed-Point). *Let* $(\mathcal{Q}, d)$ *be a complete metric space which is proved above and let* $\mathcal{T} : \mathcal{Q} \to \mathcal{Q}$ *be a contraction on* $\mathcal{Q}$ *with modulus* $\gamma$. *Then,* $\mathcal{T}$ *has a unique fixed point* $\mathbf{Q}^* \in \mathcal{Q}$ *such that* $\mathcal{T}(\mathbf{Q}) = \mathbf{Q}^*$.

**Theorem 5** (Optimal Fixed Point of Optimality Operator). *Let* $\mathbf{Q}^* \in \mathcal{Q}$ *be the optimal multi-objective value function, such that it takes multi-objective Q-value corresponding to the supremum of expected discounted rewards under a policy* $\pi$ *then* $\mathbf{Q}^* = \mathcal{T}(\mathbf{Q}^*)$.

These two theorems state that applying our multi-objective optimality operator $T$ iteratively on any multi-objective Q-value function $\mathbf{Q}$ converges to the optimal $\mathbf{Q}^*$ under metric $d$ and the Bellman equation holds at the same fixed-point. Hence, we use this optimality operator to obtain target values when optimizing the loss function in Algorithm 1.

## 4.2 PREFERENCE DRIVEN MO-DDQN-HER AND MO-TD3-HER

The main objective of this work is to obtain a single policy network to approximate the Pareto front that covers the entire preference space. For this purpose, we extend double deep Q-network (DDQN) (Van Hasselt et al., 2016) to a multi-objective version (MO-DDQN) with the preference-driven optimality operator to obtain a single parameterized function representing the $\mathcal{Q} \in \mathbb{R}^{L^{S \times A}}$ with parameters $\theta$. This network takes $s, \boldsymbol{\omega}$ as input and outputs $|\mathcal{A}| \times L$ Q-values, as described in Algorithm 1. To pull $\mathbf{Q}$ towards $T(\mathbf{Q})$, MO-DDQN minimizes the following loss function at each step $k$:

$$L_k(\theta) = \mathbb{E}_{(s,a,\mathbf{r},s',\boldsymbol{\omega}) \sim \mathcal{D}} \left[ \left( \mathbf{y} - \mathbf{Q}(s, a, \boldsymbol{\omega}; \theta) \right)^2 \right] \tag{6}$$

where $\mathcal{D}$ is the experience replay buffer that stores transitions $(s, a, \mathbf{r}, s', \boldsymbol{\omega})$ for every time step, $\mathbf{y} = \mathbf{r} + \gamma \mathbf{Q}(s', \sup_{a' \in A}(S_c(\boldsymbol{\omega}, \mathbf{Q}(s', a', \boldsymbol{\omega})) \cdot (\boldsymbol{\omega}^T \mathbf{Q}(s', a', \boldsymbol{\omega}))); \theta')$ denotes the preference-driven target value which is obtained using target Q-network's parameters $\theta'$. Here, $S_c(\boldsymbol{\omega}, \mathbf{Q}(s', a', \boldsymbol{\omega}))$ denotes the cosine similarity between the preference vector and the Q-values. Without the cosine similarity term, the supremum operator yields the action that only maximizes $\boldsymbol{\omega}^T \mathbf{Q}(s', a', \boldsymbol{\omega})$. This may pull the target Q-values in the wrong direction, especially when the scales of the objectives are in different orders of magnitude. For instance, let us assume a multi-objective problem with percent energy efficiency ($\in [0, 1]$) being the first objective and the latency ($\in [1, 10]$) in milliseconds being the second objective. Let us also assume that the preference vector ($\boldsymbol{\omega} = \{0.9, 0.1\}$) favors the energy efficiency objective and there are two separate actions ($a_1, a_2$) that yields Q-values of $Q_1 = \{0.9, 1\}, Q_2 = \{0.1, 10\}$ respectively. The supremum operator chooses action, $a_2$ since $\boldsymbol{\omega}^T Q_2 = 1.09$ is higher than $\boldsymbol{\omega}^T Q_1 = 0.91$. Since the scales of these two objectives are in different orders of magnitudes, $\sup_{a' \in A}(\boldsymbol{\omega}^T \mathbf{Q}(s', a', \boldsymbol{\omega}))$ always chooses the action that favors the latency objective. This behavior negatively affects the training since the target Q-values are pulled in the wrong direction. On the contrary, our novel cosine similarity term enables the optimality operator to choose actions that align the preferences with the Q-values and maximize the target value at the same time. The supremum operator now chooses action, $a_1$ since $S_c \cdot \boldsymbol{\omega}^T Q_1 = 0.75 \times 0.91 = 0.68$ is higher than $S_c \cdot \boldsymbol{\omega}^T Q_2 = 0.12 \times 1.09 = 0.13$. With this addition, the algorithm disregards the large $\boldsymbol{\omega}^T \mathbf{Q}(s', a', \boldsymbol{\omega})$ where the preference vector and the Q-values are not aligned since the $S_c(\boldsymbol{\omega}, \mathbf{Q}(s', a', \boldsymbol{\omega}))$ takes small values.

For each episode during training, we randomly sample a preference vector ($\boldsymbol{\omega} \in \Omega : \sum_{i=0}^{L} \omega_i = 1$) from a uniform distribution. For example, assume the randomly sampled preference vector ($\boldsymbol{\omega}$) favors energy efficiency rather than speed in multi-objective continuous control tasks. Increasing energy efficiency (i.e., lower speed) will increase the number of transitions before reaching a terminal condition since it decreases the risk of reaching a terminal state (e.g., fall). Hence, transitions that favor speed instead of energy efficiency may be underrepresented in the experience replay buffer in this example. This behavior may create a bias towards overrepresented preferences, and the network cannot learn to cover the entire preference space. To overcome this issue, we employ hindsight experience replay buffer (HER) (Andrychowicz et al., 2017), where every transition is also stored with $N_\omega$ randomly sampled preferences ($\boldsymbol{\omega}' \in \Omega : \sum_{i=0}^{L} \omega_i' = 1$) different than the original preference of the transition.

*To increase the sample efficiency of our algorithm*, we also divide the preference space into $C_p$ sub-spaces ($\tilde{\Omega}$), where $C_p$ denotes the number of child processes. Each child process is responsible for its own preference subspace. The agent, hence the network, is shared among child processes and the main process. In each child process, for each episode, we randomly sample a preference vector from child's preference sub-space ($\boldsymbol{\omega} \in \tilde{\Omega} : \sum_{i=0}^{L} \omega_i = 1$). These $C_p$ child processes run in parallel to collect transitions. These transitions are stored inside the HER in the main process. Network architectures and implementation details are available Appendix B. This parallelization provides efficient exploration during training.

**Preference alignment:** A solution in the Pareto front may not align perfectly with its preference vector. This may result in a bias in our updating scheme

---

**Algorithm 1:** Preference Driven MO-DDQN-HER

**Input:** Minibatch size $N_m$, Number of time steps $N$, Discount factor $\gamma$, Target network update coefficient $\tau$, Multi-dimensional interpolator $I(\boldsymbol{\omega})$.
**Initialize:** Replay buffer $\mathcal{D}$, Current $\mathbf{Q}_\theta$ and target network $\mathbf{Q}_{\theta'} \leftarrow \mathbf{Q}_\theta$ with parameters $\theta$ and $\theta'$.
**for** $n = 0: N$ **do**
 // Child Process
 Initialize $t = 0$ and $done = False$.
 Reset the environment to randomly initialized state $s_0$.
 Sample a preference vector $\boldsymbol{\omega}$ from the subspace $\tilde{\Omega}$.
 **while** $done = False$ **do**
  Observe state $s_t$ and select an action $a_t$ $\epsilon$-greedily:
$$a_t = \begin{cases} a \in A & w.p. \quad \epsilon \\ \max_{a \in \mathcal{A}} \boldsymbol{\omega}\mathbf{Q}(s_t, a, \boldsymbol{\omega}; \theta), & w.p. \quad 1 - \epsilon \end{cases}$$
  Observe $\mathbf{r}$, $s'$, and $done$.
  Transfer $(s_t, a_t, \mathbf{r}_t, s', \boldsymbol{\omega}, done)$ to main process.
 // Main Process
 Sample $N_\omega$ preferences $\boldsymbol{\omega}'$
 **for** $j = 1: N_\omega$ **do**
  Store transition $(s_t, a_t, \mathbf{r}_t, s', \boldsymbol{\omega}'_j, done)$ in $\mathcal{D}$
 Sample $N_m$ transitions from $\mathcal{D}$.
 $\boldsymbol{\omega}_p \leftarrow I(\boldsymbol{\omega})$
 $\mathbf{y} \leftarrow \mathbf{r} + \gamma\mathbf{Q}(s', \sup_{a' \in A}(S_c(\boldsymbol{\omega}_p, \mathbf{Q}(s', a', \boldsymbol{\omega})) \cdot$
 $(\boldsymbol{\omega}^T\mathbf{Q}(s', a', \boldsymbol{\omega})); \theta')$
 $L_k(\theta) = \mathbb{E}_{(s,a,\mathbf{r},s',\boldsymbol{\omega}) \sim \mathcal{D}}\left[\left(\mathbf{y} - \mathbf{Q}(s, a, \boldsymbol{\omega}; \theta)\right)^2\right]$
 Update $\theta_k$ by applying SGD to $L_k(\theta)$.
 Update target network parameters $\theta'_k \leftarrow \tau\theta_k + (1 - \tau)\theta'_k$

---

due to the cosine similarity term we introduced. To mitigate this adverse effect, we fit a multi-dimensional interpolator to project the original preference vectors ($\boldsymbol{\omega} \in \Omega$) to normalized solution space to align preferences with the multi-objective solutions. We identify the key preferences and obtain solutions for each of these preferences. We set key preference points as $\boldsymbol{\omega}_j = 1 : j = i, \boldsymbol{\omega}_j = 0 : j \neq i \,\forall\, i, j \in \{0, \dots, L\}$ where $i^{th}$ element corresponds to the objective we try to maximize. A preference vector of $\boldsymbol{\omega} = \{\frac{1}{L}, \dots, \frac{1}{L}\}$ is also added to this key preference set. For example, for a two-dimensional objective space, the key preference set becomes $\{[1, \ 0], [0.5, \ 0.5], [0, \ 1]\}$. We first obtain solutions for each preference in this key preference set by training the agent with a fixed preference vector and without using HER (see Figure 2(a)). We then use these solutions to obtain a normalized solution space, as shown in Figure 2(b). Here, normalization is the process of obtaining unit vectors for these solutions. For example, the normalized vector for a solution $f$ is described as $\hat{f} = \frac{f}{|f|}$. Then, we use the normalized solution space and key preference point to fit a multi-dimensional interpolator $I(\boldsymbol{\omega})$ to project the original preference vectors ($\boldsymbol{\omega} \in \Omega$) to the normalized solution space. As a result, we obtain projected preference vectors ($\boldsymbol{\omega}_p$) as illustrated in Figure 2(c). These projected preference vectors are incorporated in the cosine similarity term of the preference-driven optimality operator. This practical modification extends to theoretical results as the proof for Theorem 3 is also valid for $\boldsymbol{\omega}_p$. The interpolator is also updated during training as PD-MORL may find new non-dominated solutions for identified key preferences.

**Extension to continuous action space:** Finally, we extend the TD3 algorithm to a multi-objective version using PD-MORL for problems with continuous action space. In this case, the target values are no longer computed using the optimality operator. Instead, the actions for the target Q-value are

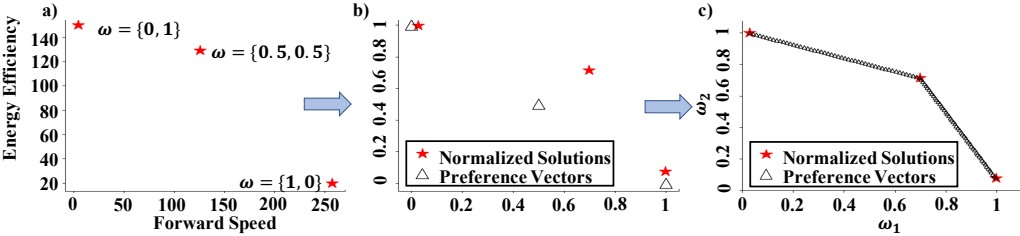

Figure 2: Overview of the interpolation scheme for the MO-Swimmer-v2 problem. (a) Obtained solutions for key preference points. (b) Normalized solution space and corresponding preference vectors. (c) Interpolated preference space.

determined by the target actor network. The actor network is updated to obtain a policy that maximizes the Q-values generated by the critic network. This update rule eventually boils down to obtaining a policy that maximizes the expected discounted rewards since, by definition, this expectation is the Q-value itself. To incorporate the relation between preference vectors and Q-values for the multi-objective version of TD3, we include a directional angle term $g(\boldsymbol{\omega}, \mathbf{Q}(s, a, \boldsymbol{\omega}; \theta))$ to both actor's and critic's loss function. This directional angle term $g(\boldsymbol{\omega}, \mathbf{Q}(s, a, \boldsymbol{\omega}; \theta)) = cos^{-1}(\frac{\boldsymbol{\omega}^T \mathbf{Q}(s,a,\boldsymbol{\omega};\theta)}{||\boldsymbol{\omega}_p|| \; ||\mathbf{Q}(s,a,\boldsymbol{\omega};\theta)||})$ denotes the angle between the preference vector ($\boldsymbol{\omega}$) and multi-objective Q-value $\mathbf{Q}(s, a, \boldsymbol{\omega}; \theta)$ and provides an alignment between preferences and the Q-values. Algorithm, network architectures, and implementation details are available in Appendix B.1.5 and B.3.

## 5 EXPERIMENTS

This section extensively evaluates the proposed PD-MORL technique using commonly used MORL benchmarks with discrete state-action spaces (Section 5.1) and complex MORL environments with continuous state-action spaces (Section 5.2). Detailed descriptions of these benchmarks are provided in Appendix B.2. Our results are compared to state-of-the-art techniques in terms of hypervolume, sparsity, and the Pareto front plots.

### 5.1 MORL BENCHMARKS WITH DISCRETE STATE-ACTION SPACES

This section illustrates the efficacy of PD-MORL in *obtaining a single universal network that covers the entire preference space* in discrete state-action environments. We use the following commonly used benchmarks to (i) make a fair comparison with prior work (Yang et al., 2019), and (ii) show that PD-MORL scales to more than two objectives:

**Deep Sea Treasure (DST)** has discrete state ($\mathcal{S} \subseteq \mathbb{N}^2$) and action spaces ($\mathcal{A} \subseteq \mathbb{N}^4$). It has two competing objectives: time penalty and treasure value. The treasure values increase as their distance from the starting point $s_0 = (0, 0)$ increases. Agent gets a -1 time penalty for every time step.

**Fruit Tree Navigation (FTN)** has discrete state ($\mathcal{S} \subseteq \mathbb{N}^2$) and action spaces ($\mathcal{A} \subseteq \mathbb{N}^2$) with six objectives: different nutrition facts of the fruits on the tree: {Protein, Carbs, Fats, Vitamins, Minerals, Water}. The goal of the agent is to find a path on the tree to collect the fruit that maximizes the nutrition facts for a given preference.

Figure 3 illustrates the policies achieved by PD-MORL for three preference vectors capture corner cases. Although our policy is *not specific* to any preference, the submarine achieves optimal treasure values for the given preference. For example, it finds the best trade-off within the shortest time one step) when we only value the time penalty ($w = \{0, 1\}$). As the importance of the treasure value increases ($w = \{0.5, 0.5\}$, $w = \{1, 0\}$), it spends optimal amount of time (8 steps and 19 steps, respectively) to find a deeper treasure. Figure 4 provides further insights into the progression of the policies found by PD-MORL. Let the first objective be the treasure value and the second objective be the time penalty in this figure. At the beginning of the training, each child process starts collecting transitions within their own preference sub-space $\tilde{\Omega}$. Since the agent acts $\epsilon$-greedily at early phases and cannot reach the higher treasure values due to terminal condition (time limit), we observe that most of the solutions are stuck in the first-subspace ($\tilde{\Omega}_1$) The transitions collected from child processes are stored in the hindsight experience replay buffer with different preferences other than their original preferences. This buffer enables extra exploration for the agent and leads the agent to different sub-spaces, as illustrated in the middle figure. As the training progresses, HER and our novel optimality operator with the angle term guide the search to expand the Pareto front and cover the entire preference space with a single network.

Among existing algorithms that learns a unified policy (Yang et al., 2019; Abels et al., 2019), the Envelope algorithm (Yang et al., 2019) is the superior and a more recent approach. Hence, we compare and evaluate PD-MORL and the Envelope algorithm on DST and FTN by obtaining a set of preference vectors that covers the entire preference space.

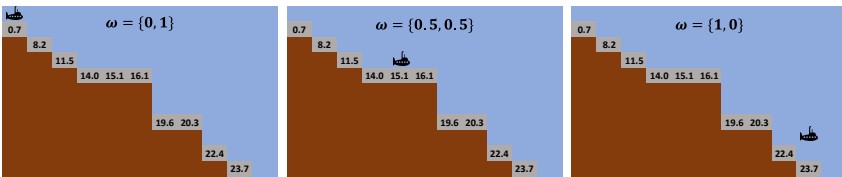

Figure 3: Optimal policies obtained by the trained agent for three corner preferences.

PD-MORL achieves both 6.1% larger hypervolume and 56% lower sparsity than the Envelope algorithm for DST problem, as shown in Table 1. Similarly, PD-MORL generates up to 78% larger hypervolume than the Envelope algorithm for FTN. Since FTN is a binary tree, the distance between Pareto solutions is the same, and hence, the

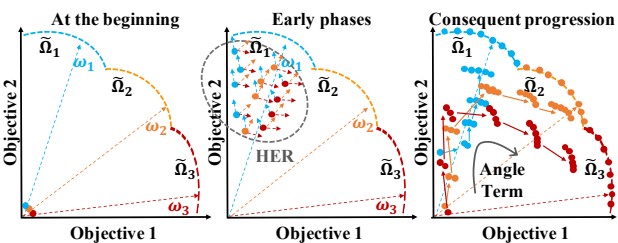

Figure 4: Policy progression during training.

sparsity metric is the same between the two approaches. Our experiments show that the improvement of PD-MORL compared to the Envelope algorithm increases with the problem complexity (i.e., increasing tree depth), as shown in Table 1. PD-MORL provides efficient and robust exploration using HER and our novel optimality operator together. Further comparisons against the Envelope algorithm (Yang et al., 2019) are provided in Appendix B.4.

Table 1: Comparison of our approach with prior work (Yang et al., 2019) using two simple MORL benchmarks in terms of hypervolume and sparsity metrics. Reference point for hypervolume calculation is set to (0,-19) and (0,0) for DST and FTN, respectively.

|  | Deep Sea Treasure | | Fruit Tree Navigation (d=6) | | Fruit Tree Navigation (d=7) | |
| --- | --- | --- | --- | --- | --- | --- |
|  | Hypervolume | Sparsity | Hypervolume | Sparsity | Hypervolume | Sparsity |
| **Envelope (Yang et al., 2019)** | 227.89 | 2.62 | 8427.51 | N/A | 6395.27 | N/A |
| **PD-MORL** | 241.73 | 1.14 | 9299.15 | N/A | 11419.58 | N/A |

## 5.2 MORL BENCHMARKS WITH CONTINUOUS CONTROL TASKS

This section evaluates PD-MORL on popular multi-objective continuous control tasks based on MuJoCo physics engine (Xu et al., 2020; Todorov et al., 2012). The environment details of these benchmarks are given in Table 2. The action and state spaces take continuous values and consist of multiple dimensions, making these benchmarks challenging to solve. The goal is to tune the amount of torque applied on the hinges/rotors for a given

Table 2: Environment details of continuous control benchmarks.

|  | State Space | Action Space |
| --- | --- | --- |
| **MO-Walker2d-v2** | $\mathcal{S} \subseteq \mathbb{R}^{17}$ | $\mathcal{A} \subseteq \mathbb{R}^{6}$ |
| **MO-HalfCheetah-v2** | $\mathcal{S} \subseteq \mathbb{R}^{17}$ | $\mathcal{A} \subseteq \mathbb{R}^{6}$ |
| **MO-Ant-v2** | $\mathcal{S} \subseteq \mathbb{R}^{27}$ | $\mathcal{A} \subseteq \mathbb{R}^{8}$ |
| **MO-Swimmer-v2** | $\mathcal{S} \subseteq \mathbb{R}^{8}$ | $\mathcal{A} \subseteq \mathbb{R}^{2}$ |
| **MO-Hopper-v2** | $\mathcal{S} \subseteq \mathbb{R}^{11}$ | $\mathcal{A} \subseteq \mathbb{R}^{3}$ |

preference $\boldsymbol{\omega}$ while satisfying multiple objectives (e.g., forward speed vs. energy efficiency, forward speed vs. jumping height, etc.). We compare the performance of PD-MORL against two state-of-the-art approaches (Xu et al., 2020; Chen et al., 2019) that use these continuous control benchmarks. We emphasize that *these approaches must learn a different policy network for every solution in the Pareto front. In contrast, PD-MORL learns a single universal network that covers the entire preference space. The Envelope algorithm (Yang et al., 2019) is not included in these comparisons since it was not evaluated with continuous action spaces, and PD-MORL already outperforms it on simpler problems.* Training details are reported in Appendix B.3.

We first compare PD-MORL to prior work using hypervolume and sparsity metrics. Since META (Chen et al., 2019) and PG-MORL (Xu et al., 2020) report the average of six runs, we also ran each benchmark six times with PD-MORL. The average metrics are reported in Table 3, while the standard deviations and results of individual runs are given in Appendix B.4 Table 8.

The desired Pareto front approximation should have high hypervolume and low sparsity metrics. PD-MORL outperforms the current state-of-the-art techniques both in terms of hypervolume and sparsity on every benchmark except MO-Hopper-v2, as summarized in Table 3. We emphasize that our *PD-MORL technique trains only a single network, while the other methods use customized policies network for each Pareto point.* For example, we achieve 12% higher hypervolume and 25% better sparsity than the most competitive prior work PG-MORL (Xu et al., 2020) on the Walker environment. They reported 263 Pareto front solutions for this environment. This corresponds to, in total, $2.8 \times 10^6$ trainable parameters since they have different policy network for each solution. In contrast, PD-MORL achieves better or comparable result using only $3.4 \times 10^5$ trainable parameters. This may enable the deployment of PD-MORL in a real-life scenario where there are dynamic changes in the design constraints and goals. We also note that PG-MORL achieves better sparsity metric for MO-HalfCheetah-v2 and MO-Ant-v2 environments. However, PG-MORL also achieves a smaller hypervolume compared to PD-MORL. This suggests that to determine the quality of a Pareto front,

Table 3: Performance comparison of the proposed approach and state-of-the-art algorithms on the continuous control benchmarks in terms of hypervolume and sparsity metrics. Reference point for hypervolume calculation is set to (0,0) point. HV*: Hypervolume

| | MO-Walker2d-v2 | | MO-HalfCheetah-v2 | | MO-Ant-v2 | | MO-Swimmer-v2 | | MO-Hopper-v2 | |
| --- | --- | --- | --- | --- | --- | --- | --- | --- | --- | --- |
| | HV* | Sparsity | HV* | Sparsity | HV* | Sparsity | HV* | Sparsity | HV* | Sparsity |
| PG-MORL (Xu et al., 2020) | $4.82 \times 10^6$ | $0.04 \times 10^4$ | $5.77 \times 10^6$ | $\mathbf{0.44 \times 10^3}$ | $6.35 \times 10^6$ | $\mathbf{0.37 \times 10^4}$ | $2.57 \times 10^4$ | 9.9 | $\mathbf{2.02 \times 10^7}$ | $0.5 \times 10^4$ |
| META (Chen et al., 2019) | $2.10 \times 10^6$ | $2.10 \times 10^4$ | $5.18 \times 10^6$ | $2.13 \times 10^3$ | $2.40 \times 10^4$ | $1.56 \times 10^4$ | $1.23 \times 10^4$ | 24.4 | $1.25 \times 10^7$ | $4.84 \times 10^4$ |
| **PD-MORL (Ours)** | $\mathbf{5.41 \times 10^6}$ | $\mathbf{0.03 \times 10^4}$ | $\mathbf{5.89 \times 10^6}$ | $0.49 \times 10^3$ | $\mathbf{7.48 \times 10^6}$ | $0.78 \times 10^4$ | $\mathbf{3.21 \times 10^4}$ | 5.7 | $1.88 \times 10^7$ | $\mathbf{0.3 \times 10^4}$ |

these metrics are rather limited and should be further investigated by the existing literature. Therefore, we also plot Pareto front plots for all algorithms in Figure 5(a)-(e) to have a better understanding of the quality of the Pareto front. Figure 5 shows that the proposed PD-MORL technique achieves a broader and denser Pareto front than the state-of-the-art approaches despite using a single unified network. PD-MORL mostly relies on the interpolation procedure before the actual training. The key solutions obtained during this period determine the initial convergence of the algorithm. However, for MO-Hopper-v2, the obtained key solutions for preference vectors $\{1, 0\}, \{0.5, 0.5\}, \{0, 1\}$ are $\{1778, 4971\}, \{1227, 1310\}, \{3533, 3165\}$ respectively. These key solutions are not a good representative of the Pareto front shown in Figure 5(e). To have representative key solutions is a limitation of PD-MORL; however, it can be solved with hyperparameter tuning.

Finally, Figure 5(f) plots the progression of the hypervolume of the Pareto front for the environments with similar scales. The plots show that PD-MORL effectively pushes the hypervolume by discovering new Pareto solutions with the help of efficient and robust exploration. We ensure this by using HER, our novel directional angle term, and dividing the preference space into sub-spaces to collect transitions in parallel. The progression of the Pareto front, sparsity, and hypervolume of all benchmarks are provided in Appendix B.4.

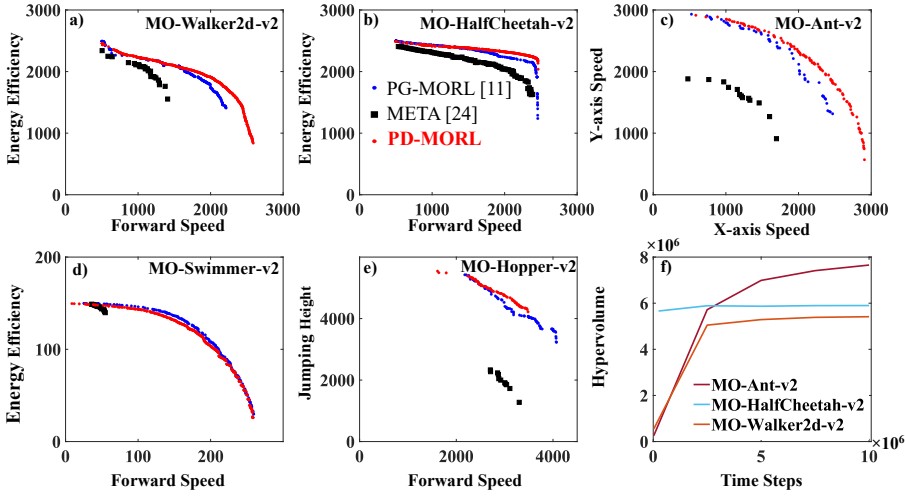

Figure 5: Pareto front comparison for continuous control tasks (a)-(e). Results for META and PG-MORL are obtained from (Xu et al., 2020) and code base of PG-MORL. (f) Progression of hypervolume of the Pareto front. Results for environments that have similar scales are given.

# 6 CONCLUSIONS AND LIMITATIONS

Many real-life applications have multiple conflicting objectives that can change dynamically. Existing MORL techniques either learn a different policy for each preference or fail to extend to complex continuous state-action benchmarks. This paper presented a novel MORL technique that addresses both of these limitations. It trains a universal network to cover the entire preference space using the preferences as guidance to update the network, hence the name preference-driven MORL (PD-MORL). We evaluated PD-MORL using both discrete and continuous benchmarks and demonstrated that PD-MORL outperforms the state-of-the-art approaches.

PD-MORL aims to obtain a generalized network that covers the entire preference space. It updates its network parameters by sampling transitions from HER. Since these transitions can represent anywhere in the preference space, the updates should be conservative. Therefore, it may fail to obtain a dense Pareto front on problems with massive action, state, and preference spaces which is the primary concern of any MORL algorithm.

ACKNOWLEDGMENTS

This work was supported in part by NSF CAREER award CNS-1651624, and DARPA Young Faculty Award (YFA) Grant D14AP00068.

ETHICS STATEMENT

This work introduces a novel multi-objective reinforcement learning approach scalable to continuous control tasks. While reinforcement learning has several ethical consequences, especially in settings where the automated decision affects humans directly, our algorithmic approach does not address these concerns specifically. Therefore, we do not expect any ethical issues raised by our work. Instead, it helps to alleviate these concerns since considering multiple objectives in societal problems enables various trade-offs.

REPRODUCIBILITY STATEMENT

Proofs of all our theoretical results are given in Appendix A. Appendix B provides details of the implementation of our approach and additional experimental results. The source code is attached with the rest of the supplementary material, providing a complete description of the multi-objective RL environments and instructions on reproducing our experiments.

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

# Supplementary Material: PD-MORL: Preference-Driven Multi-Objective Reinforcement Learning Algorithm

## A    THEORETICAL ANALYSIS OF THE DIRECTIONAL ANGLE GUIDED MORL ALGORITHM

This section introduces a theoretical analysis of the proposed PD-MORL algorithm. We follow the theoretical framework for value-based MORL algorithms proposed by Yang et al. (2019), which is based on Banach's Fixed-Point Theorem (also known as Contraction Mapping Theorem). This theorem states that every *contraction* on a *complete* metric space has a unique fixed-point. Considering this, we define (i) contraction operators and (ii) a metric space to design our value-based MORL algorithm.

### A.1    METRIC SPACE

In standard Q-learning, the value space is defined as $\mathcal{Q} \in \mathbb{R}^{\mathcal{S} \times \mathcal{A}}$, containing all bounded functions $Q(s, a)$ which are the estimates of the total expected rewards when the agent is at state $s$, taking action $a$. We extend it to a multi-objective value space by defining the value space as $\mathcal{Q} \in \mathbb{R}^{L^{\mathcal{S} \times \mathcal{A}}}$, containing all bounded functions $\mathbf{Q}(s, a, \boldsymbol{\omega})$ which are the estimates of expected total rewards under preference $\boldsymbol{\omega} \in \mathbb{R}^{L}_{\geq 0} : \sum_{i=0}^{L} \omega_i = 1$. We then define a metric in this value space as:

$$d(\mathbf{Q}, \mathbf{Q}') := \sup_{s \in \mathcal{S}, a \in \mathcal{A}, \boldsymbol{\omega} \in \Omega} |\boldsymbol{\omega}^T(\mathbf{Q}(s, a, \boldsymbol{\omega}) - \mathbf{Q}'(s, a, \boldsymbol{\omega}))|. \tag{7}$$

This metric gives the distance between $\mathbf{Q}$ and $\mathbf{Q}'$ as the supremum norm of the scalarized distance between these two vectors. Notice that $d$ should satisfy the following axioms to be considered as a metric:

- **Non-negativity**: $d(\mathbf{Q}, \mathbf{Q}') \geq 0$. This axiom holds for our metric $d$. The definition of supremum norm is the largest value of a set of absolute values, and thus, it is greater than or equal to zero.

- **Symmetry**: $d(\mathbf{Q}, \mathbf{Q}') = d(\mathbf{Q}', \mathbf{Q})$. Similarly to above axiom, this axiom also holds for $d$ since absolute values are considered in supremum norm.

- **Triangle Inequality**: $d(\mathbf{Q}, \mathbf{Q}') \leq d(\mathbf{Q}, \mathbf{Q}'') + d(\mathbf{Q}'', \mathbf{Q}')$. This axiom holds for metric $d$ as shown by the following proof:

$$\sup_{\substack{s \in \mathcal{S}, a \in \mathcal{A} \\ \boldsymbol{\omega} \in \Omega}} |\boldsymbol{\omega}^T(\mathbf{Q}(s, a, \boldsymbol{\omega}) - \mathbf{Q}''(s, a, \boldsymbol{\omega}))| + \sup_{\substack{s \in \mathcal{S}, a \in \mathcal{A} \\ \boldsymbol{\omega} \in \Omega}} |\boldsymbol{\omega}^T(\mathbf{Q}''(s, a, \boldsymbol{\omega}) - \mathbf{Q}'(s, a, \boldsymbol{\omega}))|$$

$$\geq |\boldsymbol{\omega}^T(\mathbf{Q}(s, a, \boldsymbol{\omega}) - \mathbf{Q}''(s, a, \boldsymbol{\omega}))| + |\boldsymbol{\omega}^T(\mathbf{Q}''(s, a, \boldsymbol{\omega}) - \mathbf{Q}'(s, a, \boldsymbol{\omega}))|$$

$$\geq |\boldsymbol{\omega}^T(\mathbf{Q}(s, a, \boldsymbol{\omega}) - \mathbf{Q}'(s, a, \boldsymbol{\omega}))|$$

$\sup_{\substack{s \in \mathcal{S}, a \in \mathcal{A} \\ \boldsymbol{\omega} \in \Omega}} |\boldsymbol{\omega}^T(\mathbf{Q}(s, a, \boldsymbol{\omega}) \; - \; \mathbf{Q}''(s, a, \boldsymbol{\omega}))| \; + \; \sup_{\substack{s \in \mathcal{S}, a \in \mathcal{A} \\ \boldsymbol{\omega} \in \Omega}} |\boldsymbol{\omega}^T(\mathbf{Q}''(s, a, \boldsymbol{\omega}) \; - \; \mathbf{Q}'(s, a, \boldsymbol{\omega}))|$ is the upper bound on the set $|\boldsymbol{\omega}^T(\mathbf{Q}(s, a, \boldsymbol{\omega}) - \mathbf{Q}'(s, a, \boldsymbol{\omega}))|$. Hence,

by definition of supremum norm (least upper bound):

$$\sup_{\substack{s\in\mathcal{S},a\in\mathcal{A}\\\boldsymbol{\omega}\in\Omega}}|\boldsymbol{\omega}^T(\mathbf{Q}(s,a,\boldsymbol{\omega})-\mathbf{Q}'(s,a,\boldsymbol{\omega}))| \leq \sup_{\substack{s\in\mathcal{S},a\in\mathcal{A}\\\boldsymbol{\omega}\in\Omega}}|\boldsymbol{\omega}^T(\mathbf{Q}(s,a,\boldsymbol{\omega})-\mathbf{Q}''(s,a,\boldsymbol{\omega}))|$$
$$+ \sup_{\substack{s\in\mathcal{S},a\in\mathcal{A}\\\boldsymbol{\omega}\in\Omega}}|\boldsymbol{\omega}^T(\mathbf{Q}''(s,a,\boldsymbol{\omega})-\mathbf{Q}'(s,a,\boldsymbol{\omega}))|$$

- **Identity of indiscernibles**: $d(\mathbf{Q},\mathbf{Q}')=0 \Leftrightarrow \mathbf{Q}=\mathbf{Q}'$. This axiom does not hold for metric $d$ since the dot product of different $\mathbf{Q}(s,a,\boldsymbol{\omega})$ functions may result in same value for a specific preference vector $\boldsymbol{\omega}$.

In summary, the metric $d$ is a pseudo-metric since the identity of indiscernibles does not hold for it.

In the following Theorem, we showed that the metric space equipped with this metric is complete since the limit point of the sequence of operators is required to be in this space.

**Theorem 1** (Multi-objective Metric Space $(\mathcal{Q},d)$ is Complete)**.** *The metric space $(\mathcal{Q},d)$ is complete and every Cauchy sequence $\mathbf{Q}_n(s,a,\boldsymbol{\omega})$ is convergent in metric space $(\mathcal{Q},d)$ $\forall\,s,a,\boldsymbol{\omega}\in\mathcal{S},\mathcal{A},\Omega$.*

*Proof.* Let $\mathbf{Q}_n(s,a,\boldsymbol{\omega})$ be a Cauchy sequence in $\mathcal{Q}$. Given $\epsilon > 0$, there exist $n,m \geq N > 0$ such that $\sup_{s\in\mathcal{S},a\in\mathcal{A},\boldsymbol{\omega}\in\Omega}|\boldsymbol{\omega}^T(\mathbf{Q}_n(s,a,\boldsymbol{\omega})-\mathbf{Q}_m(s,a,\boldsymbol{\omega}))| < \epsilon$. Hence,

$$|\boldsymbol{\omega}^T(\mathbf{Q}_n(s,a,\boldsymbol{\omega})-\mathbf{Q}_m(s,a,\boldsymbol{\omega}))| \leq \sup_{\substack{s\in\mathcal{S}\\a\in\mathcal{A}\\\boldsymbol{\omega}\in\Omega}}|\boldsymbol{\omega}^T(\mathbf{Q}_n(s,a,\boldsymbol{\omega})-\mathbf{Q}_m(s,a,\boldsymbol{\omega}))| < \epsilon,\,\forall\,s,a,\boldsymbol{\omega}\in\mathcal{S},\mathcal{A},\Omega$$

This implies that for each $s,a,\boldsymbol{\omega}\in\mathcal{S},\mathcal{A},\Omega$, the sequence of L-dimensional real numbers $\mathbf{Q}_n(s,a,\boldsymbol{\omega})$ is a Cauchy sequence. Since $\mathbb{R}^L$ is complete, $\mathbf{Q}_n(s,a,\boldsymbol{\omega})$ is convergent. Let $\mathbf{Q}(s,a,\boldsymbol{\omega}) = \lim_{n\to\infty}\mathbf{Q}_n(s,a,\boldsymbol{\omega})$. Cauchy sequence in a normed space must be bounded. Let there be an $M > 0$ such that $|\boldsymbol{\omega}^T(\mathbf{Q}_n(s,a,\boldsymbol{\omega}))| \leq \sup_{s\in\mathcal{S},a\in\mathcal{A},\boldsymbol{\omega}\in\Omega}|\boldsymbol{\omega}^T(\mathbf{Q}_n(s,a,\boldsymbol{\omega}))| \leq M,\,\forall\,s\in\mathcal{S},a\in\mathcal{A},\boldsymbol{\omega}\in\Omega$. Taking $n\to\infty$, we find $|\boldsymbol{\omega}^T(\mathbf{Q}(s,a,\boldsymbol{\omega}))| = \lim_{n\to\infty}\boldsymbol{\omega}^T(\mathbf{Q}_n(s,a,\boldsymbol{\omega})) \leq M,\,\forall\,s\in\mathcal{S},a\in\mathcal{A},\boldsymbol{\omega}\in\Omega$. This shows that $\mathbf{Q}(s,a,\boldsymbol{\omega})$ is a bounded function and thus $\mathbf{Q}\in\mathcal{Q}$. For all $n\geq N$,

$$|\boldsymbol{\omega}^T(\mathbf{Q}_n(s,a,\boldsymbol{\omega})-\mathbf{Q}(s,a,\boldsymbol{\omega}))| = \lim_{m\to\infty}|\boldsymbol{\omega}^T(\mathbf{Q}_n(s,a,\boldsymbol{\omega})-\mathbf{Q}_m(s,a,\boldsymbol{\omega}))| \leq \epsilon,\,\forall\,s,a,\boldsymbol{\omega}\in\mathcal{S},\mathcal{A},\Omega$$

and hence $\forall\,n\geq N, \sup_{s\in\mathcal{S},a\in\mathcal{A},\boldsymbol{\omega}\in\Omega}|\boldsymbol{\omega}^T(\mathbf{Q}_n(s,a,\boldsymbol{\omega})-\mathbf{Q}(s,a,\boldsymbol{\omega}))| \leq \epsilon$. This implies that $\mathbf{Q}(s,a,\boldsymbol{\omega})$ is the limit of $\mathbf{Q}_n(s,a,\boldsymbol{\omega})$ and proves that $\mathbf{Q}_n(s,a,\boldsymbol{\omega})$ is convergent in $\mathcal{Q}$.

This completes our proof that the metric space $(\mathcal{Q},d)$ is complete. $\qquad\square$

### A.2 Multi-Objective Bellman's Evaluation and Preference-Driven Optimality Operators

Given a policy $\pi$ and sampled transition $\tau$, we can define multi-objective Bellman's evaluation operator $\mathcal{T}_\pi$ using the metric space $(\mathcal{Q},d)$ as:

$$(\mathcal{T}_\pi\mathbf{Q})(s,a,\boldsymbol{\omega}) := \mathbf{r}(s,a) + \gamma\mathbb{E}_{\mathcal{T}\sim(\mathcal{P},\pi)}\mathbf{Q}(s',a',\boldsymbol{\omega}) \tag{8}$$

where $(s',a',\boldsymbol{\omega})$ denotes the next state-action-preference pair and $\gamma\in(0,1)$ is the discount factor.

**Theorem 2** (Multi-objective Bellman's Evaluation Operator is Contraction)**.** *Let $(\mathcal{Q},d)$ be a complete metric space (as in Theorem 1). Let $\mathbf{Q}$ and $\mathbf{Q}'$ be any two multi-objective Q-value functions in this space. The multi-objective Bellman's evaluation operator is a contraction and $d(\mathcal{T}_\pi\mathbf{Q},\mathcal{T}_\pi\mathbf{Q}') \leq \gamma d(\mathbf{Q},\mathbf{Q}')$ holds for the Lipschitz constant $\gamma$ (the discount factor).*

*Proof.* We start by expanding the expression $d(\mathcal{T}_\pi\mathbf{Q},\mathcal{T}_\pi\mathbf{Q}')$:

$$d(\mathcal{T}_\pi\mathbf{Q},\mathcal{T}_\pi\mathbf{Q}') = \sup_{\substack{s\in\mathcal{S},a\in\mathcal{A}\\\boldsymbol{\omega}\in\Omega}}|\boldsymbol{\omega}^T(\mathcal{T}_\pi\mathbf{Q}(s,a,\boldsymbol{\omega})-\mathcal{T}_\pi\mathbf{Q}'(s,a,\boldsymbol{\omega}))|$$
$$= \sup_{\boldsymbol{\omega}\in\Omega}|\gamma\boldsymbol{\omega}^T\mathbb{E}_{\mathcal{T}\sim(\mathcal{P},\pi)}\mathbf{Q}(s',a',\boldsymbol{\omega}) - \gamma\boldsymbol{\omega}^T\mathbb{E}_{\mathcal{T}\sim(\mathcal{P},\pi)}\mathbf{Q}'(s',a',\boldsymbol{\omega})|$$

$$= \gamma \cdot \sup_{\boldsymbol{\omega} \in \Omega} |\boldsymbol{\omega}^T \mathbb{E}_{\mathcal{T} \sim (\mathcal{P}, \pi)} (\mathbf{Q}(s', a', \boldsymbol{\omega}) - \mathbf{Q}'(s', a', \boldsymbol{\omega}))|$$

$$\leq \gamma \cdot \sup_{\boldsymbol{\omega} \in \Omega} \boldsymbol{\omega}^T \mathbb{E}_{\mathcal{T} \sim (\mathcal{P}, \pi)} |\mathbf{Q}(s', a', \boldsymbol{\omega}) - \mathbf{Q}'(s', a', \boldsymbol{\omega})| \quad (|\mathbb{E}[\cdot]| \leq \mathbb{E}[|\cdot|])$$

$$= \gamma \cdot \sup_{\boldsymbol{\omega} \in \Omega} \mathbb{E}_{\mathcal{T} \sim (\mathcal{P}, \pi)} |\boldsymbol{\omega}^T (\mathbf{Q}(s', a', \boldsymbol{\omega}) - \mathbf{Q}'(s', a', \boldsymbol{\omega}))|$$

$$\leq \gamma \cdot \sup_{\boldsymbol{\omega} \in \Omega} \sup_{\substack{s' \in \mathcal{S} \\ a' \in \mathcal{A} \\ \boldsymbol{\omega} \in \Omega}} |\boldsymbol{\omega}^T (\mathbf{Q}(s', a', \boldsymbol{\omega}) - \mathbf{Q}'(s', a', \boldsymbol{\omega}))| \quad (\mathbb{E}[|\cdot|] \leq \sup |\cdot|)$$

$$= \gamma \cdot \sup_{\substack{s' \in \mathcal{S} \\ a' \in \mathcal{A} \\ \boldsymbol{\omega} \in \Omega}} |\boldsymbol{\omega}^T (\mathbf{Q}(s', a', \boldsymbol{\omega}) - \mathbf{Q}'(s', a', \boldsymbol{\omega}))| = \gamma \cdot d(\mathbf{Q}, \mathbf{Q}')$$

$$d(\mathcal{T}_\pi \mathbf{Q}, \mathcal{T}_\pi \mathbf{Q}') \leq \gamma \cdot d(\mathbf{Q}, \mathbf{Q}')$$

To avoid confusion, the last step (Step 7) has only one supremum norm since the outside supremum does not see any variable. The definition of the variables ends with the inside supremum. This completes our proof that multi-objective Bellman's evaluation operator, $\mathcal{T}_\pi$, is contraction. □

We define a *preference-driven optimality operator* $\mathcal{T}$ by adding a cosine similarity term between preference vectors and state-action values to the multi-objective Bellman's optimality operator as:

$$(\mathcal{T}\mathbf{Q})(s, a, \boldsymbol{\omega}) := \mathbf{r}(s, a) + \gamma \mathbb{E}_{s' \sim \mathcal{P}(\cdot|s,a)} \mathbf{Q}(s', \sup_{a' \in \mathcal{A}} (S_c(\boldsymbol{\omega}, \mathbf{Q}(s', a', \boldsymbol{\omega})) \cdot (\boldsymbol{\omega}^T \mathbf{Q}(s', a', \boldsymbol{\omega}))), \boldsymbol{\omega})$$

(9)

where $S_c(\boldsymbol{\omega}, \mathbf{Q}(s', a', \boldsymbol{\omega}))$ denotes the cosine similarity between preference vector and Q-value. $\sup_{a' \in \mathcal{A}} (S_c(\boldsymbol{\omega}, \mathbf{Q}(s', a', \boldsymbol{\omega})) \cdot (\boldsymbol{\omega}^T \mathbf{Q}(s', a', \boldsymbol{\omega})))$ yields the action $a'$ that maximizes the multiplication inside the supremum. This term enables our optimality operator to choose actions that align the preferences with the Q-values and maximize the target value, as elaborated in Section 4.2 under the preference alignment subtitle.

**Theorem 3** (Preference-driven Multi-objective Bellman's Optimality Operator is Contraction). *Let* $(\mathcal{Q}, d)$ *be a complete metric space. Let* $\mathbf{Q}$ *and* $\mathbf{Q}'$ *be any two multi-objective Q-value functions in this space. The preference-driven multi-objective Bellman's optimality operator is a contraction and* $d(\mathcal{T}\mathbf{Q}, \mathcal{T}\mathbf{Q}') \leq \gamma d(\mathbf{Q}, \mathbf{Q}')$ *holds for the Lipschitz constant* $\gamma$ *(the discount factor).*

*Proof.* We use $S_c(\boldsymbol{\omega}, \mathbf{Q}(s', a', \boldsymbol{\omega}))$ as $S_c$ and $S_c(\boldsymbol{\omega}, \mathbf{Q}'(s', a'', \boldsymbol{\omega}))$ as $S_c'$ in the proof for notational simplicity. We start by expanding the expression $d(\mathcal{T}\mathbf{Q}, \mathcal{T}\mathbf{Q}')$:

$$d(\mathcal{T}\mathbf{Q}, \mathcal{T}\mathbf{Q}') = \sup_{\substack{s \in \mathcal{S}, a \in \mathcal{A} \\ \boldsymbol{\omega} \in \Omega}} |\boldsymbol{\omega}^T (\mathcal{T}\mathbf{Q}(s, a, \boldsymbol{\omega}) - \mathcal{T}\mathbf{Q}'(s, a, \boldsymbol{\omega}))|$$

$$= \sup_{\substack{s \in \mathcal{S}, a \in \mathcal{A} \\ \boldsymbol{\omega} \in \Omega}} \left| \gamma \boldsymbol{\omega}^T \mathbb{E}_{s' \sim \mathcal{P}(\cdot|s,a)} \mathbf{Q}(s', \sup_{a' \in \mathcal{A}} (S_c) \cdot (\boldsymbol{\omega}^T \mathbf{Q}(s', a', \boldsymbol{\omega})), \boldsymbol{\omega}) \right.$$
$$\left. - \gamma \boldsymbol{\omega}^T \mathbb{E}_{s' \sim \mathcal{P}(\cdot|s,a)} \mathbf{Q}'(s', \sup_{a'' \in \mathcal{A}} (S_c') \cdot (\boldsymbol{\omega}^T \mathbf{Q}'(s', a'', \boldsymbol{\omega})), \boldsymbol{\omega}) \right|$$

$$= \gamma \cdot \sup_{\substack{s \in \mathcal{S}, a \in \mathcal{A} \\ \boldsymbol{\omega} \in \Omega}} \left| \mathbb{E}_{s' \sim \mathcal{P}(\cdot|s,a)} \left[ \boldsymbol{\omega}^T \Big( \mathbf{Q}(s', \sup_{a' \in \mathcal{A}} (S_c) \cdot (\boldsymbol{\omega}^T \mathbf{Q}(s', a', \boldsymbol{\omega})), \boldsymbol{\omega}) \right. \right.$$
$$\left. \left. - \mathbf{Q}'(s', \sup_{a'' \in \mathcal{A}} (S_c') \cdot (\boldsymbol{\omega}^T \mathbf{Q}'(s', a'', \boldsymbol{\omega})), \boldsymbol{\omega}) \Big) \right] \right|$$

$$\leq \gamma \cdot \sup_{\substack{s \in \mathcal{S}, a \in \mathcal{A} \\ \boldsymbol{\omega} \in \Omega}} \mathbb{E}_{s' \sim \mathcal{P}(\cdot|s,a)} \left[ \left| \boldsymbol{\omega}^T \Big( \mathbf{Q}(s', \sup_{a' \in \mathcal{A}} (S_c) \cdot (\boldsymbol{\omega}^T \mathbf{Q}(s', a', \boldsymbol{\omega})), \boldsymbol{\omega}) \right. \right.$$
$$\left. \left. - \mathbf{Q}'(s', \sup_{a'' \in \mathcal{A}} (S_c') \cdot (\boldsymbol{\omega}^T \mathbf{Q}'(s', a'', \boldsymbol{\omega})), \boldsymbol{\omega}) \Big) \right| \right] \quad (|\mathbb{E}[\cdot]| \leq \mathbb{E}[|\cdot|])$$

$$\leq \gamma \cdot \sup_{\substack{s \in \mathcal{S}, a \in \mathcal{A} \\ \boldsymbol{\omega} \in \Omega}} \sup_{s' \in \mathcal{S}, \boldsymbol{\omega} \in \Omega} \left| \boldsymbol{\omega}^T \Big( \mathbf{Q}(s', \sup_{a' \in \mathcal{A}} (S_c) \cdot (\boldsymbol{\omega}^T \mathbf{Q}(s', a', \boldsymbol{\omega})), \boldsymbol{\omega}) \right.$$

$$- \mathbf{Q}'(s', \sup_{a'' \in \mathcal{A}} (S'_c) \cdot (\boldsymbol{\omega}^T \mathbf{Q}'(s', a'', \boldsymbol{\omega})), \boldsymbol{\omega}))\Big| \quad (\mathbb{E}[|\cdot|] \leq \sup|\cdot|)$$

$$= \gamma \cdot \sup_{s' \in \mathcal{S}, \boldsymbol{\omega} \in \Omega} \Big| \boldsymbol{\omega}^T \Big( \mathbf{Q}(s', \sup_{a' \in \mathcal{A}} (S_c) \cdot (\boldsymbol{\omega}^T \mathbf{Q}(s', a', \boldsymbol{\omega})), \boldsymbol{\omega})$$

$$- \mathbf{Q}'(s', \sup_{a'' \in \mathcal{A}} (S'_c) \cdot (\boldsymbol{\omega}^T \mathbf{Q}'(s', a'', \boldsymbol{\omega})), \boldsymbol{\omega}))\Big| \quad (Rearrange\ supremums)$$

Let $a'$ be the action that maximizes $(S_c) \cdot (\boldsymbol{\omega}^T \mathbf{Q}(s', a', \boldsymbol{\omega}))$ for state $s'$ and preference $\boldsymbol{\omega}$, then we have :

$$d(\mathcal{T}\mathbf{Q}, \mathcal{T}\mathbf{Q}') \leq \gamma \cdot \sup_{s' \in \mathcal{S}, \boldsymbol{\omega} \in \Omega} \Big| \boldsymbol{\omega}^T \Big( \mathbf{Q}(s', a', \boldsymbol{\omega}) - \mathbf{Q}'(s', \sup_{a'' \in \mathcal{A}} (S'_c) \cdot (\boldsymbol{\omega}^T \mathbf{Q}'(s', a'', \boldsymbol{\omega})), \boldsymbol{\omega})) \Big) \Big|$$

W.l.o.g we assume $\boldsymbol{\omega}^T \mathbf{Q}(s', a', \boldsymbol{\omega}) - \boldsymbol{\omega}^T \mathbf{Q}'(s', \sup_{a'' \in \mathcal{A}} (S'_c) \cdot (\boldsymbol{\omega}^T \mathbf{Q}'(s', a'', \boldsymbol{\omega})), \boldsymbol{\omega}) \geq 0$. Proof is similar for the other inequality case. Since $\boldsymbol{\omega}^T \mathbf{Q}'(s', a', \boldsymbol{\omega}) \leq \boldsymbol{\omega}^T \mathbf{Q}'(s', \sup_{a'' \in \mathcal{A}} (S'_c) \cdot (\boldsymbol{\omega}^T \mathbf{Q}'(s', a'', \boldsymbol{\omega})), \boldsymbol{\omega})$, we have:

$$d(\mathcal{T}\mathbf{Q}, \mathcal{T}\mathbf{Q}') \leq \gamma \cdot \sup_{\substack{s' \in \mathcal{S}, a \in \mathcal{A} \\ \boldsymbol{\omega} \in \Omega}} \Big| \boldsymbol{\omega}^T \Big( \mathbf{Q}(s', a', \boldsymbol{\omega}) - \mathbf{Q}'(s', a', \boldsymbol{\omega}) \Big) \Big| = \gamma \cdot d(\mathbf{Q}, \mathbf{Q}')$$

$$d(\mathcal{T}\mathbf{Q}, \mathcal{T}\mathbf{Q}') \leq \gamma \cdot d(\mathbf{Q}, \mathbf{Q}')$$

This completes our proof that preference-driven multi-objective Bellman's optimality operator, $\mathcal{T}$, is contraction. $\qquad \square$

Theorem 2 and Theorem 3 state that multi-objective evaluation and optimality operators are contractions. They ensure that we can apply our optimality operator in Equation 9 iteratively to obtain the optimal multi-objective value function given by Theorem 4 and Theorem 5 below, respectively.

**Theorem 4** (Preference-driven Multi-objective Optimality Operator Converges to a Fixed-Point). *Let $(\mathcal{Q}, d)$ be a complete metric space which is proved above and let $\mathcal{T} : \mathcal{Q} \to \mathcal{Q}$ be a contraction on $\mathcal{Q}$ with modulus $\gamma$. Then, $\mathcal{T}$ has a unique fixed-point $\mathbf{Q}^* \in \mathcal{Q}$ such that $\mathcal{T}(\mathbf{Q}) = \mathbf{Q}^*$.*

*Proof.* Let $\mathbf{Q}_0 \in \mathcal{Q}$ and define a sequence $(\mathbf{Q}_n)$ where $\mathbf{Q}_{n+1} = \mathcal{T}(\mathbf{Q}_n), n = 1, 2, \ldots$

$$\begin{aligned} d(\mathbf{Q}_{n+1}, \mathbf{Q}_n) &= d(\mathcal{T}(\mathbf{Q}_n), \mathcal{T}(\mathbf{Q}_{n-1})) \\ &\leq \gamma d(\mathbf{Q}_n, \mathbf{Q}_{n-1}) = \gamma d(\mathcal{T}(\mathbf{Q}_{n-1}), \mathcal{T}(\mathbf{Q}_{n-2})) \\ &\leq \gamma^2 d(\mathbf{Q}_{n-1}, \mathbf{Q}_{n-2}) \\ &\vdots \\ &\leq \gamma^m d(\mathbf{Q}_1, \mathbf{Q}_0) \end{aligned}$$

Hence, for $m > n \ \forall \ m, n \in \mathbb{N}$ by the triangle inequality we have

$$\begin{aligned} d(\mathbf{Q}_m, \mathbf{Q}_n) &\leq d(\mathbf{Q}_m, \mathbf{Q}_{m-1}) + d(\mathbf{Q}_{m-1}, \mathbf{Q}_{m-2}) + \ldots + d(\mathbf{Q}_{n+1}, \mathbf{Q}_n) \\ &\leq (\gamma^{m-1} + \gamma^{m-2} + \ldots + \gamma^n) d(\mathbf{Q}_1, \mathbf{Q}_0) \\ &\leq \frac{\gamma^n}{1 - \gamma} d(\mathbf{Q}_1, \mathbf{Q}_0) \end{aligned}$$

Therefore, $\mathbf{Q}_n$ is Cauchy: for $\epsilon > 0$, let $N$ be large enough that $\frac{\gamma^n}{1-\gamma} d(\mathbf{Q}_1, \mathbf{Q}_0) < \epsilon$, which ensures that $n, m > N \Rightarrow d(\mathbf{Q}_n, \mathbf{Q}_m) < \epsilon$. Since $(\mathcal{Q}, d)$ is complete, $\mathbf{Q}_n$ converges to $\mathbf{Q}^* \in \mathcal{Q}$. Now, we show that $\mathbf{Q}^*$ is indeed a fixed point since $\mathbf{Q}_n$ converges to $\mathbf{Q}^*$ and $\mathcal{T}$ is continuous.

$$\mathbf{Q}^* = \lim_{n \to \infty} \mathbf{Q}_n = \lim_{n \to \infty} \mathcal{T}(\mathbf{Q}_{n-1}) = \mathcal{T}(\lim_{n \to \infty} \mathbf{Q}_{n-1}) = \mathcal{T}(\mathbf{Q}^*)$$

Finally, $\mathcal{T}$ cannot have more than one fixed point in $(\mathcal{Q}, d)$, since any pair of distinct fixed points $\mathbf{Q}_1^*$ and $\mathbf{Q}_2^*$ would contradict the contraction of $\mathcal{T}$:

$$d(\mathcal{T}(\mathbf{Q}_1^*), \mathcal{T}(\mathbf{Q}_2^*)) = d(\mathbf{Q}_1^*, \mathbf{Q}_2^*) > \gamma d(\mathbf{Q}_1^*, \mathbf{Q}_2^*)$$

This completes our proof that multi-objective optimality operator converges to $\mathbf{Q}^*$. $\qquad\square$

**Theorem 5** (Optimal Fixed Point of Optimality Operator). *Let $\mathbf{Q}^* \in \mathcal{Q}$ be the optimal multi-objective value function, such that it takes multi-objective Q-value corresponding to the supremum of expected discounted rewards under a policy $\pi$ then $\mathbf{Q}^* = \mathcal{T}(\mathbf{Q}^*)$.*

*Proof.* We start by defining the optimal Q-function, $\mathbf{Q}^*$ as:

$$\mathbf{Q}^*(s, a, \boldsymbol{\omega}) = \arg_{\mathbf{Q}} \sup_{\pi \in \Pi} \boldsymbol{\omega}^T \mathbb{E}_{\mathcal{T} \sim (\mathcal{P}, \pi)|s_0=s, a_0=a} \Big[ \sum_{t=0}^{\infty} \gamma^t \mathbf{r}(s_t, a_t) \Big].$$

From Theorem 4 we observe that $\lim_{n \to \infty} d(\mathcal{T}^n(\mathbf{Q}), \mathbf{Q}^*) = 0$ for any $\mathbf{Q} \in \mathcal{Q}$. This suggests that $\boldsymbol{\omega}^T \mathcal{T}(\mathbf{Q}^*)(s, a, \boldsymbol{\omega}) = \boldsymbol{\omega}^T \mathbf{Q}^*(s, a, \boldsymbol{\omega})$ from the definition of metric $d$. Expanding the $\boldsymbol{\omega}^T \mathcal{T}(\mathbf{Q}^*)(s, a, \boldsymbol{\omega})$, we have:

$$\boldsymbol{\omega}^T \mathcal{T}(\mathbf{Q}^*)(s, a, \boldsymbol{\omega}) = \boldsymbol{\omega}^T \mathbf{r}(s, a) + \gamma \cdot \boldsymbol{\omega}^T \mathbb{E}_{s' \sim \mathcal{P}(\cdot|s,a)} \mathbf{Q}^*(s', \sup_{a' \in \mathcal{A}} (S_c(\boldsymbol{\omega}, \mathbf{Q}^*(s', a', \boldsymbol{\omega})) \cdot (\boldsymbol{\omega}^T \mathbf{Q}^*(s', a', \boldsymbol{\omega}))), \boldsymbol{\omega}).$$

Let $a'$ is the action that maximizes the $\sup_{a' \in \mathcal{A}}(S_c(\boldsymbol{\omega}, \mathbf{Q}^*(s', a', \boldsymbol{\omega})) \cdot (\boldsymbol{\omega}^T \mathbf{Q}^*(s', a', \boldsymbol{\omega}))$ for state $s'$ and preference $\boldsymbol{\omega}$. Then, by substituting $Q^*$ into $\mathcal{T}(\mathbf{Q}^*)$ we have:

$$\boldsymbol{\omega}^T \mathcal{T}(\mathbf{Q}^*)(s, a, \boldsymbol{\omega}) = \boldsymbol{\omega}^T \mathbf{r}(s, a) + \gamma \cdot \boldsymbol{\omega}^T \mathbb{E}_{s' \sim \mathcal{P}(\cdot|s,a)} \Big[ \arg_{\mathbf{Q}} \sup_{\pi \in \Pi} \boldsymbol{\omega}^T \mathbb{E}_{\mathcal{T} \sim (\mathcal{P}, \pi)|s_0=s', a_0=a'} \Big[ \sum_{t=0}^{\infty} \gamma^t \mathbf{r}(s_t, a_t) \Big] \Big]$$

$$= \boldsymbol{\omega}^T \mathbf{r}(s, a) + \gamma \cdot \boldsymbol{\omega}^T \arg_{\mathbf{Q}} \sup_{\pi \in \Pi} \boldsymbol{\omega}^T \mathbb{E}_{\substack{\mathcal{T} \sim (\mathcal{P}, \pi) \\ s_0 \sim \mathcal{P}(\cdot|s,a)}} \sum_{t=0}^{\infty} \gamma^t \mathbf{r}(s_t, a_t)$$

Now, we merge $\mathbf{r}(s, a)$ with the summation operation by rearranging the conditions of the expectation operator and obtain:

$$= \boldsymbol{\omega}^T \Big( \arg_{\mathbf{Q}} \sup_{\pi \in \Pi} \boldsymbol{\omega}^T \mathbb{E}_{\mathcal{T} \sim (\mathcal{P}, \pi)|s_0=s, a_0=a} \Big[ \sum_{t=0}^{\infty} \gamma^t \mathbf{r}(s_t, a_t) \Big] \Big) = \boldsymbol{\omega}^T \mathbf{Q}^*(s, a, \boldsymbol{\omega})$$

$$\boldsymbol{\omega}^T \mathcal{T}(\mathbf{Q}^*)(s, a, \boldsymbol{\omega}) = \boldsymbol{\omega}^T \mathbf{Q}^*(s, a, \boldsymbol{\omega})$$

This completes our proof that optimal multi-objective value function $\mathbf{Q}^*$ is the fixed point of the preference-driven multi-objective optimality operator $\mathcal{T}$. $\qquad\square$

# B  IMPLEMENTATION-TRAINING DETAILS AND ADDITIONAL EXPERIMENTAL RESULTS

This section provides details on the Preference-Driven MO-DDQN-HER and MO-TD3-HER algorithms. It expands the explanation of the algorithms with implementation details. Then, it explains the training details for all benchmarks. Furthermore, we provide additional experimental results for discrete MORL and multi-objective continuous control benchmarks.

## B.1  PREFERENCE-DRIVEN MO-DDQN-HER AND MO-TD3-HER

### B.1.1  HINDSIGHT EXPERIENCE REPLAY

We randomly sample a preference vector ($\boldsymbol{\omega} \in \Omega : \sum_{i=0}^{L} \omega_i = 1$) from a uniform distribution in each episode during training. Depending on the problem, the number of transitions observed by interacting with the environment for some preferences may be underrepresented. This behavior creates a bias towards overrepresented preferences, and the network cannot learn to cover the entire preference

space. Therefore, we employ hindsight experience replay buffer (HER) (Andrychowicz et al., 2017), where every transition is also stored with $N_\omega$ randomly sampled preferences ($\omega' \in \Omega : \sum_{i=0}^{L} \omega_i' = 1$) from a uniform distribution different than the original preference of the transition. Specifically, for each transition ($s, a, \mathbf{r}, s', \omega, done$), additional $N_\omega$ transitions ($s, a, \mathbf{r}, s', \omega', done$) are also stored in the experience replay buffer. This strategy provides efficient exploration and generalizability to the agent. Using HER, the agent learns to recover from undesired states and continue to align itself with its original preference.

### B.1.2 EXPLORATION IN PARALLEL

To further increase the sample efficiency of our algorithm, we equally divide the preference space into $C_p$ sub-spaces ($\tilde{\Omega}$), where $C_p$ denotes the number of child processes. The number of child processes $C_p$ is set to 10 for all benchmarks. The agent is shared among these child processes and the main process. In each child process, for each episode, we randomly sample a preference vector from child's preference sub-space ($\omega \in \tilde{\Omega} : \sum_{i=0}^{L} \omega_i = 1$). These $C_p$ child processes run in parallel to collect transitions. After collecting a transition, each child process transfers it to the main process and waits for other child processes. After the main process receives transitions from every child process and stores them using HER, the child processes continue to collect transitions. Since we collect and store $C_p$ transitions at every child-main process loop, networks are also updated $C_p$ times. Parallel exploration and HER together provides efficient exploration.

### B.1.3 PREFERENCE ALIGNMENT WITH INTERPOLATION

A solution in the Pareto front may not align perfectly with its preference vector. To mitigate this adverse effect, we fit a multi-dimensional interpolator to project the original preference vectors ($\omega \in \Omega$) to normalized solution space to align preferences with the multi-objective solutions. Here, normalization is the process of obtaining unit vectors for these solutions. For example, the normalized vector for a solution $f$ is described as $\hat{f} = \frac{f}{|f|}$. We identify the key preferences and obtain solutions for each of these preferences without HER. We set key preference points as $\omega_j = 1 : j = i, \omega_j = 0 : j \neq i \,\forall\, i, j \in \{0, \ldots, L\}$ where $i^{th}$ element corresponds to the objective we try to maximize. A preference vector of $\omega = \{\frac{1}{L}, \ldots, \frac{1}{L}\}$ is also added to this key preference set. We first obtain solutions for each preference in this key preference set by training the agent with a fixed preference vector and without using HER. We use these solutions first to obtain a normalized solution space. Then, we use the normalized solution space and key preference vectors to fit a multi-dimensional interpolator $I(\omega)$ to project the original preference vectors ($\omega \in \Omega$) to the normalized solution space. We use radial basis function interpolation with linear kernel in this work (Virtanen et al., 2020). As a result, we obtain projected preference vectors ($\omega_p$) that are incorporated in the cosine similarity term of the preference-driven optimality operator for the MO-DDQN-HER algorithm and in the directional angle term for the MO-TD3-HER algorithm. The interpolator is also updated during training as PD-MORL may find new non-dominated solutions for identified key preferences. To this end, at every episode, we evaluate our agent using the key preference set.

### B.1.4 DETAILS ON MO-DDQN-HER

The main objective of this work is to obtain a single policy network to approximate the Pareto front that covers the entire preference space. For this purpose, we extend double deep Q-network (DDQN) (Van Hasselt et al., 2016) to a multi-objective version (MO-DDQN) with the preference-driven optimality operator. Algorithm 2 describes the training using the proposed MO-DDQN-HER.

We first initialize an empty buffer $\mathcal{D}$, Q-network and target Q-network with parameters $\theta$ and $\theta'$. The preference space is then equally divided into $C_p$ sub-spaces, where we initialize a child process for each sub-space. In each child process, for each episode, we randomly sample a preference vector from child's preference sub-space ($\omega \in \tilde{\Omega} : \sum_{i=0}^{L} \omega_i = 1$). The agent interacts with the environment using $\epsilon$-greedy policy and collect transition ($s, a, \mathbf{r}, s', \omega, done$). This transition is then transferred to the main process. After main process receives transitions from every child process, it stores transitions inside $\mathcal{D}$. For each transition, we also store $N_\omega$ transitions where a different preference ($\omega' \in \Omega : \sum_{i=0}^{L} \omega_i' = 1$) is sampled. Specifically, for each transition ($s, a, \mathbf{r}, s', \omega, done$), additional $N_\omega$ transitions ($s, a, \mathbf{r}, s', \omega', done$) are also stored in $\mathcal{D}$. We then sample a minibatch of

transitions from $\mathcal{D}$ to update the network. To calculate the cosine similarity metric, we first project the preferences $\boldsymbol{\omega}$ to normalized solution space and obtain projected preferences $\boldsymbol{\omega}_p$. MO-DDQN-HER minimizes the following loss function at each step $k$:

$$L_k(\theta) = \mathbb{E}_{(s,a,\mathbf{r},s',\boldsymbol{\omega})\sim\mathcal{D}}\left[\left(\mathbf{y} - \mathbf{Q}(s,a,\boldsymbol{\omega};\theta)\right)^2\right] \tag{10}$$

where $\mathbf{y} = \mathbf{r} + \gamma\mathbf{Q}(s', \sup_{a'\in A}(S_c(\boldsymbol{\omega}, \mathbf{Q}(s',a',\boldsymbol{\omega}))\cdot(\boldsymbol{\omega}^T\mathbf{Q}(s',a',\boldsymbol{\omega})));\theta')$ denotes the preference-driven target value which is obtained using target Q-network's parameters $\theta'$. Instead of MSE, Smooth L1 loss may also be used for this loss function. We soft update the target network at every time step $k$. Note that each child process runs for $N$ time steps. Hence, in total, we collect $N \times C_p$ transitions.

---

**Algorithm 2:** Preference Driven MO-DDQN-HER

---

**Input:** Minibatch size $N_m$, Number of time steps $N$,
Discount factor $\gamma$, Target network update coefficient $\tau$,
Multi-dimensional interpolator $I(\boldsymbol{\omega})$.
**Initialize:** Replay buffer $\mathcal{D}$, Current $\mathbf{Q}_\theta$ and target network $\mathbf{Q}_{\theta'} \leftarrow \mathbf{Q}_\theta$ with parameters $\theta$ and $\theta'$.
**for** $n$ = 0: $N$ **do**
    // Child Process
    Initialize $t = 0$ and $done = False$.
    Reset the environment to randomly initialized state $s_0$.
    Sample a preference vector $\boldsymbol{\omega}$ from the subspace $\tilde{\Omega}$.
    **while** $done = False$ **do**
        Observe state $s_t$ and select an action $a_t$ $\epsilon$-greedily:
$$a_t = \begin{cases} a \in A & w.p. \quad \epsilon \\ \max_{a\in\mathcal{A}} \boldsymbol{\omega}\mathbf{Q}(s_t,a,\boldsymbol{\omega};\theta), & w.p. \quad 1-\epsilon \end{cases}$$
        Observe $\mathbf{r}$, $s'$, and $done$.
        Transfer $(s_t, a_t, \mathbf{r}_t, s', \boldsymbol{\omega}, done)$ to main process.
    // Main Process
    Store the transition $(s_t, a_t, \mathbf{r}_t, s', \boldsymbol{\omega}, done)$ obtained from every child process in $\mathcal{D}$.
    Sample $N_\omega$ preferences $\boldsymbol{\omega}'$
    **for** $j$ = 1: $N_\omega$ **do**
        Store transition $(s_t, a_t, \mathbf{r}_t, s', \boldsymbol{\omega}'_j, done)$ in $\mathcal{D}$
    Sample $N_m$ transitions from $\mathcal{D}$.
    $\boldsymbol{\omega}_p \leftarrow I(\boldsymbol{\omega})$
    $\mathbf{y} \leftarrow \mathbf{r} + \gamma\mathbf{Q}(s', \sup_{a'\in A}(S_c(\boldsymbol{\omega}_p, \mathbf{Q}(s',a',\boldsymbol{\omega}))\cdot(\boldsymbol{\omega}^T\mathbf{Q}(s',a',\boldsymbol{\omega}));\theta')$
    $L_k(\theta) = \mathbb{E}_{(s,a,\mathbf{r},s',\boldsymbol{\omega})\sim\mathcal{D}}\left[\left(\mathbf{y} - \mathbf{Q}(s,a,\boldsymbol{\omega};\theta)\right)^2\right]$
    Update $\theta_k$ by applying SGD to $L_k(\theta)$.
    Update target network parameters $\theta'_k \leftarrow \tau\theta_k + (1-\tau)\theta'_k$

---

### B.1.5 DETAILS ON MO-TD3-HER

We extend the TD3 algorithm to a multi-objective version using PD-MORL for problems with continuous action space. To incorporate the relation between preference vectors and Q-values for the multi-objective version of TD3, we include a directional angle term $g(\boldsymbol{\omega}, \mathbf{Q}(s,a,\boldsymbol{\omega};\theta))$ to both actor's and critic's loss function. This directional angle term $g(\boldsymbol{\omega}, \mathbf{Q}(s,a,\boldsymbol{\omega};\theta)) = cos^{-1}(\frac{\boldsymbol{\omega}^T\mathbf{Q}(s,a,\boldsymbol{\omega};\theta)}{||\boldsymbol{\omega}_p|| \, ||\mathbf{Q}(s,a,\boldsymbol{\omega};\theta)||})$ denotes the angle between the preference vector ($\boldsymbol{\omega}$) and multi-objective Q-value $\mathbf{Q}(s,a,\boldsymbol{\omega};\theta)$ and provides an alignment between preferences and the Q-values.

Similar to MO-DDQN-HER, we first initialize an empty buffer $\mathcal{D}$, critic networks $\mathbf{Q}_{\theta_1}, \mathbf{Q}_{\theta_2}$ and actor network $\pi_\phi$ with parameters $\theta_1$, $\theta_2$, and $\phi$. We also initialize critic and actor target networks. The process of initialization of child processes is the same with MO-DDQN-HER. The agent interacts with the environment to collect transition $(s,a,\mathbf{r},s',\boldsymbol{\omega}, done)$ by selecting actions according to a policy with an exploration noise term $\epsilon$. The utilization of HER for collected transitions is also the same as the MO-DDQN-HER. We then sample a minibatch of transitions from $\mathcal{D}$. The algorithm computes actions for the next state using the target actor network plus a target smoothing noise. In the conventional TD3 algorithm, both critics use a single target value to update their parameters calculated using whichever of the two critics gives a smaller $Q$-value. In multi-objective version of TD3, we modified this clipped double-Q-learning approach such that target $\mathbf{Q}$-value is calculated

using whichever of the two critics gives a smaller $\boldsymbol{\omega}^T\mathbf{Q}$. To calculate the directional angle term, similar to MO-DDQN-HER, we project the preferences $\boldsymbol{\omega}$ to normalized solution space and obtain projected preferences $\boldsymbol{\omega}_p$. MO-TD3-HER minimizes the following loss function to update both critic networks at each step $k$:

$$L_{critic_k}(\theta_i) = \mathbb{E}_{(s,a,\mathbf{r},s',\boldsymbol{\omega})\sim\mathcal{D}}\Big[\big(\mathbf{y} - \mathbf{Q}(s,a,\boldsymbol{\omega};\theta_i)\big)^2\Big] + \mathbb{E}_{(s,a,\boldsymbol{\omega})\sim\mathcal{D}}\Big[g(\boldsymbol{\omega}_p, \mathbf{Q}(s,a,\boldsymbol{\omega};\theta_i))\Big] \quad (11)$$

where $\mathbf{y} = \mathbf{r} + \gamma \arg_{\mathbf{Q}} \min_{i=1,2} \boldsymbol{\omega}^T\mathbf{Q}(s',\tilde{a},\boldsymbol{\omega};\theta_i')$ denotes the target value which is obtained using target critic network's parameters. Instead of MSE, Smooth L1 loss may also be used for this loss function. One of the essential tricks that TD3 has is that it updates actor and target networks less frequently than the critics. The actor network is updated every $p_{delay}$ step by maximizing the $\boldsymbol{\omega}^T\mathbf{Q}$ while minimizing the directional angle term using the following loss:

$$\nabla_\phi L_{actor_k}(\phi) = \mathbb{E}_{(s,a,\mathbf{r},s',\boldsymbol{\omega})\sim\mathcal{D}}\Big[\nabla_a \boldsymbol{\omega}^T\mathbf{Q}(s,a,\boldsymbol{\omega};\theta_1)|_{a=\pi(s,\boldsymbol{\omega};\phi)}\nabla_\phi\pi(s,\boldsymbol{\omega};\phi)\Big] +$$
$$\alpha \cdot \mathbb{E}_{(s,a,\boldsymbol{\omega})\sim\mathcal{D}}\Big[\nabla_a g(\boldsymbol{\omega}_p, \mathbf{Q}(s,a,\boldsymbol{\omega};\theta_1)|_{a=\pi(s,\boldsymbol{\omega};\phi)}\nabla_\phi\pi(s,\boldsymbol{\omega};\phi)\Big] \quad (12)$$

where $\alpha$ denotes the loss coefficient to scale up the directional angle term to match the $\boldsymbol{\omega}^T\mathbf{Q}$ term. Here, we also soft update the target networks at every time step $k$. Similar to MO-DDQN-HER, each child process runs for $N$ time steps. Hence, in total, we collect $N \times C_p$ transitions during training.

---

**Algorithm 3:** Preference-Driven MO-TD3-HER

---

**Input:** Minibatch size $N_m$, Number of time steps $N$
Discount factor $\gamma$, Target network update coefficient $\tau$,
Multi-dimensional interpolator $I(\boldsymbol{\omega})$,
Policy update delay $p_{delay}$, Standard deviation for Gaussian exploration noise added to the policy $\sigma$,
Standard deviation for smoothing noise added to target policy $\sigma'$,
Limit for absolute value of target policy smoothing noise $c$.
**Initialize:** Replay buffer $\mathcal{D}$, Critic networks $\mathbf{Q}_{\theta_1},\mathbf{Q}_{\theta_2}$ and actor network $\pi_\phi$ with parameters $\theta_1, \theta_2$, and $\phi$,
Target networks $\mathbf{Q}_{\theta_1'} \leftarrow \mathbf{Q}_{\theta_1}$, $\mathbf{Q}_{\theta_2'} \leftarrow \mathbf{Q}_{\theta_2}$, $\pi_{\phi'} \leftarrow \pi_\phi$,
**for** $n = 0$: $N$ **do**
    // Child Process
    Initialize $t = 0$ and $done = False$.
    Reset the environment to randomly initialized state $s_0$.
    Sample a preference vector $\boldsymbol{\omega}$ from the subspace $\tilde{\Omega}$
    **while** $done = False$ **do**
        Observe state $s_t$ and select an action with exploration noise $a_t \sim \pi(s_t,\boldsymbol{\omega};\phi) + \epsilon : \epsilon \sim \mathcal{N}(0,\sigma)$
        Observe reward $\mathbf{r}$, $s'$, and $done$.
        Transfer $(s_t, a_t, \mathbf{r}_t, s', \boldsymbol{\omega}, done)$ to main process.
    // Main Process
    Store the transition $(s_t, a_t, \mathbf{r}_t, s', \boldsymbol{\omega}, done)$ obtained from every child process in $\mathcal{D}$
    Sample $N_\omega$ preferences $\boldsymbol{\omega}'$
    **for** $j = 1$: $N_\omega$ **do**
        Store transition $(s_t, a_t, \mathbf{r}_t, s', \boldsymbol{\omega}'_j, done)$ in $\mathcal{D}$
    Sample $N_m$ transitions from $\mathcal{D}$.
    $\tilde{a} \leftarrow \pi(s',\boldsymbol{\omega};\phi') + \epsilon : \epsilon \sim clip(\mathcal{N}(0,\sigma'), -c, c)$
    $\mathbf{y} \leftarrow \mathbf{r} + \gamma \arg_{\mathbf{Q}} \min_{i=1,2} \boldsymbol{\omega}^T\mathbf{Q}(s',\tilde{a},\boldsymbol{\omega};\theta_i')$
    $\boldsymbol{\omega}_p \leftarrow I(\boldsymbol{\omega})$
    $g(\boldsymbol{\omega}_p, \mathbf{Q}(s,a,\boldsymbol{\omega};\theta_i)) \leftarrow cos^{-1}\big(\frac{\boldsymbol{\omega}_p^T\mathbf{Q}(s,a,\boldsymbol{\omega};\theta_i)}{||\boldsymbol{\omega}_p|| \, ||\mathbf{Q}(s,a,\boldsymbol{\omega};\theta_i)||}\big)$
    $L_{critic_k}(\theta_i) = \mathbb{E}_{(s,a,\mathbf{r},s',\boldsymbol{\omega})\sim\mathcal{D}}\Big[\big(\mathbf{y} - \mathbf{Q}(s,a,\boldsymbol{\omega};\theta_i)\big)^2\Big] + \mathbb{E}_{(s,a,\boldsymbol{\omega})\sim\mathcal{D}}\Big[g(\boldsymbol{\omega}_p, \mathbf{Q}(s,a,\boldsymbol{\omega};\theta_i))\Big]$
    Update $\theta_{i_k}$ by applying SGD to $L_{critic_k}(\theta_i)$.
    **if** $n \bmod p_{delay}$ **then**
        $\nabla_\phi L_{actor_k}(\phi) = \mathbb{E}_{(s,a,\mathbf{r},s',\boldsymbol{\omega})\sim\mathcal{D}}\Big[\nabla_a \boldsymbol{\omega}^T\mathbf{Q}(s,a,\boldsymbol{\omega};\theta_1)|_{a=\pi(s,\boldsymbol{\omega};\phi)}\nabla_\phi\pi(s,\boldsymbol{\omega};\phi)\Big] +$
        $\alpha \cdot \mathbb{E}_{(s,a,\boldsymbol{\omega})\sim\mathcal{D}}\Big[\nabla_a g(\boldsymbol{\omega}_p, \mathbf{Q}(s,a,\boldsymbol{\omega};\theta_1)|_{a=\pi(s,\boldsymbol{\omega};\phi)}\nabla_\phi\pi(s,\boldsymbol{\omega};\phi)\Big]$
        Update target critics parameters $\theta_{i_k}' \leftarrow \tau\theta_{i_k} + (1-\tau)\theta_{i_k}'$
        Update target actor parameters $\phi_k' \leftarrow \tau\phi_k + (1-\tau)\phi_k'$

---

## B.2 Benchmarks

We first evaluate PD-MORL's performance on two commonly used discrete MORL benchmarks: Deep Sea Treasure and Fruit Tree Navigation. For these benchmarks, we make use of the codebase provided by Yang et al. (2019). We further evaluate PD-MORL on multi-objective continuous control tasks such as MO-Walker2d-v2, MO-HalfCheetah-v2, MO-Ant-v2, MO-Swimmer-v2, and MO-Hopper-v2. These benchmarks are presented by Xu et al. (2020) and are licensed under the terms of the MIT license. The details for all benchmarks are provided below:

**Deep Sea Treasure (DST):** A well-studied MORL benchmark(Hayes et al., 2022; Liu et al., 2014) where an agent is a submarine trying to collect treasures in a $10 \times 11$ grid-world. The treasure values increase as their distance from the starting point $s_0 = (0, 0)$ increase. The submarine has two objectives: the time penalty and the treasure value. The actions are navigation in four directions and are discrete. The reward is a two-element vector. The first element shows the treasure value, and the second element shows the time penalty.

**Fruit Tree Navigation (FTN):** A recent MORL benchmark presented by Yang et al. (2019). It is a binary tree of depth $d$ with randomly assigned reward $\mathbf{r} \in \mathbb{R}^6$ on the leaf nodes. These rewards show the amounts of six different nutrition facts of the fruits on the tree: $\{Protein, Carbs, Fats, Vitamins, Minerals, Water\}$. The goal of the agent is to find a path on the tree to collect the fruit that maximizes the nutrition facts for a given preference.

**MO-Walker2d-v2:** The state space and action space is defined as $\mathcal{S} \subseteq \mathbb{R}^{17}, \mathcal{A} \subseteq \mathbb{R}^6$. The agent is a two-dimensional two-legged figure. It has two objectives to consider: forward speed and energy efficiency. The goal is to tune the amount of torque applied on the hinges for a given preference $\boldsymbol{\omega}$ while moving in the forward direction.

**MO-HalfCheetah-v2:** The state space and action space is defined as $\mathcal{S} \subseteq \mathbb{R}^{17}, \mathcal{A} \subseteq \mathbb{R}^6$. The agent is a two-dimensional robot that resembles a cheetah. It has two objectives to consider: forward speed and energy efficiency. The goal is to tune the amount of torque applied on the joints for a given preference $\boldsymbol{\omega}$ while running in the forward direction.

**MO-Ant-v2:** The state space and action space is defined as $\mathcal{S} \subseteq \mathbb{R}^{27}, \mathcal{A} \subseteq \mathbb{R}^8$. The agent is a 3D robot that resembles an ant. It has two objectives to consider: x-axis speed and y-axis speed. The goal is to tune the amount of torque applied on the hinges connecting the legs and the torso for a given preference $\boldsymbol{\omega}$.

**MO-Swimmer-v2:** The state space and action space is defined as $\mathcal{S} \subseteq \mathbb{R}^8, \mathcal{A} \subseteq \mathbb{R}^2$. The agent is a two-dimensional robot. It has two objectives to consider: forward speed and energy efficiency. The goal is to tune the amount of torque applied on the rotors for a given preference $\boldsymbol{\omega}$ while swimming in the forward direction inside a two-dimensional pool.

**MO-Hopper-v2:** The state space and action space is defined as $\mathcal{S} \subseteq \mathbb{R}^{11}, \mathcal{A} \subseteq \mathbb{R}^3$. The agent is a two-dimensional one-legged figure. It has two objectives to consider: forward speed and jumping height. The goal is to tune the amount of torque applied on the hinges for a given preference $\boldsymbol{\omega}$ while moving in the forward direction by making hops.

## B.3 Training Details

We run all our experiments on a local server including Intel Xeon Gold 6242R. We do not use any GPU in our implementation. For Deep Sea Treasure and Fruit Tree Navigation benchmarks, we employ the proposed MO-DDQN-HER algorithm. The network here takes state $s$ and preference $\boldsymbol{\omega}$ as inputs and outputs $|\mathcal{A}| \times L$ Q-values. The number of hidden layers and hidden neurons among other hyperparameters for MO-DDQN-HER are given in Table 4.

For continuous control benchmarks, we employ the proposed MO-TD3-HER algorithm. The critic network takes state $s$, preference $\boldsymbol{\omega}$, and action $a$ as input and outputs $L$ Q-values for each objective. The actor network takes state $s$ and preference $\boldsymbol{\omega}$ as inputs and outputs $|\mathcal{A}|$ actions. The number of hidden layers and hidden neurons among other hyperparameters for MO-TD3-HER for each continuous benchmark are given in Table 5. The hyperparameters are the same for all benchmarks except the policy update delay for MO-Hopper-v2, which is set to 20.

As it is elaborated in Section B.1.3, we first obtain key solutions for each of the preferences in the key preference set to fit a multi-dimensional interpolator. Since we already know the Pareto front solutions for discrete benchmarks, we directly use key solutions in the Pareto front to fit this interpolator. For continuous control benchmarks, we train an agent utilizing the multi-objective version of TD3 with a fixed key preference vector and without using HER. The number of hidden layers and hidden neurons among other hyperparameters for this approach on each continuous benchmark are given in Table 6.

Table 4: Hyperparameters for MO-DDQN-HER

|  | Deep Sea Treasure | Fruit Tree Navigation |
|---|---|---|
| **Total number of steps** $(N)$ | $1 \times 10^5$ | $1 \times 10^5$ |
| **Minibatch size** $(N_m)$ | 32 | 32 |
| **Discount factor** $(\gamma)$ | 0.99 | 0.99 |
| **Soft update coefficient** $(\tau)$ | 0.005 | 0.005 |
| **Buffer size** | $1 \times 10^4$ | $1 \times 10^4$ |
| **Number of child processes** $(C_p)$ | 10 | 10 |
| **Number of preferences sampled for HER** $(N_\omega)$ | 3 | 3 |
| **Learning rate** | $3 \times 10^{-4}$ | $3 \times 10^{-4}$ |
| **Number of hidden layers** | 3 | 3 |
| **Number of hidden neurons** | 256 | 512 |

Table 5: Hyperparameters for MO-TD3-HER

|  | MO-Walker2d-v2 | MO-HalfCheetah-v2 | MO-Ant-v2 | MO-Swimmer-v2 | MO-Hopper-v2 |
|---|---|---|---|---|---|
| **Total number of steps** | $1 \times 10^6$ | $1 \times 10^6$ | $1 \times 10^6$ | $1 \times 10^6$ | $1 \times 10^6$ |
| **Minibatch size** | 256 | 256 | 256 | 256 | 256 |
| **Discount factor** | 0.995 | 0.995 | 0.995 | 0.995 | 0.995 |
| **Soft update coefficient** | 0.005 | 0.005 | 0.005 | 0.005 | 0.005 |
| **Buffer size** | $2 \times 10^6$ | $2 \times 10^6$ | $2 \times 10^6$ | $2 \times 10^6$ | $2 \times 10^6$ |
| **Number of child processes** | 10 | 10 | 10 | 10 | 10 |
| **Number of preferences sampled for HER** | 3 | 3 | 3 | 3 | 3 |
| **Learning rate-Critic** | $3 \times 10^{-4}$ | $3 \times 10^{-4}$ | $3 \times 10^{-4}$ | $3 \times 10^{-4}$ | $3 \times 10^{-4}$ |
| **Number of hidden layers - Critic** | 1 | 1 | 1 | 1 | 1 |
| **Number of hidden neurons - Critic** | 400 | 400 | 400 | 400 | 400 |
| **Learning rate-Actor** | $3 \times 10^{-4}$ | $3 \times 10^{-4}$ | $3 \times 10^{-4}$ | $3 \times 10^{-4}$ | $3 \times 10^{-4}$ |
| **Number of hidden layers - Actor** | 1 | 1 | 1 | 1 | 1 |
| **Number of hidden neurons - Actor** | 400 | 400 | 400 | 400 | 400 |
| **Policy update delay** | 10 | 10 | 10 | 10 | 20 |
| **Exploration noise std.** | 0.1 | 0.1 | 0.1 | 0.1 | 0.1 |
| **Target policy smoothing noise std.** | 0.2 | 0.2 | 0.2 | 0.2 | 0.2 |
| **Noise clipping limit** | 0.5 | 0.5 | 0.5 | 0.5 | 0.5 |
| **Loss coefficient** | 10 | 10 | 10 | 10 | 10 |

Table 6: Hyperparameters for MO-TD3 algorithm to obtain key solutions

|  | MO-Walker2d-v2 | MO-HalfCheetah-v2 | MO-Ant-v2 | MO-Swimmer-v2 | MO-Hopper-v2 |
|---|---|---|---|---|---|
| **Total number of steps** | $2 \times 10^6$ | $2 \times 10^6$ | $1 \times 10^6$ | $1 \times 10^6$ | $1 \times 10^6$ |
| **Minibatch size** | 100 | 100 | 100 | 100 | 100 |
| **Discount factor** | 0.99 | 0.99 | 0.99 | 0.99 | 0.99 |
| **Soft update coefficient** | 0.005 | 0.005 | 0.005 | 0.005 | 0.005 |
| **Buffer size** | $5 \times 10^5$ | $5 \times 10^5$ | $5 \times 10^5$ | $1 \times 10^6$ | $1 \times 10^6$ |
| **Learning rate-Critic** | $3 \times 10^{-4}$ | $3 \times 10^{-4}$ | $3 \times 10^{-4}$ | $3 \times 10^{-4}$ | $3 \times 10^{-4}$ |
| **Number of hidden layers - Critic** | 1 | 1 | 1 | 1 | 1 |
| **Number of hidden neurons - Critic** | 400 | 400 | 400 | 400 | 400 |
| **Learning rate-Actor** | $3 \times 10^{-4}$ | $3 \times 10^{-4}$ | $3 \times 10^{-4}$ | $3 \times 10^{-4}$ | $3 \times 10^{-4}$ |
| **Number of hidden layers - Actor** | 1 | 1 | 1 | 1 | 1 |
| **Number of hidden neurons - Actor** | 400 | 400 | 400 | 400 | 400 |
| **Policy update delay** | 2 | 2 | 2 | 5 | 10 |
| **Exploration noise std.** | 0.1 | 0.1 | 0.1 | 0.1 | 0.1 |
| **Target policy smoothing noise std.** | 0.2 | 0.2 | 0.2 | 0.2 | 0.2 |
| **Noise clipping limit** | 0.5 | 0.5 | 0.5 | 0.5 | 0.5 |

## B.4 Additional Experimental Results

Since the main objective of this work is to obtain a single universal network that covers the entire preference space, we first obtain a representative set of preference vectors. For Deep Sea Treasure benchmark, we obtain a preference vector set with a step size of 0.01 ($\{0, 1\}, \{0.01, 0.99\}, \ldots, \{0.99, 0.01\}, \{1, 0\}$) Similarly, we obtain preference vector sets with a step size of 0.1 and 0.001 for Fruit Tree Navigation and continuous control benchmarks, respectively.

Since the Envelope algorithm (Yang et al., 2019) is the superior and more recent approach that learns a unified policy, we first compare PD-MORL with this algorithm using discrete MORL benchmarks. For the Envelope algorithm, we use the authors' codebase with their default hyperparameters mentioned

in the paper since we believe that they are already optimized for these two benchmarks. In addition to hypervolume and sparsity metrics that we provide in Section 5.1, we also compare PD-MORL with the Envelope algorithm using the coverage ration F1 (CRF1) metric. This metric, which is coined by Yang et al. (2019), evaluates the agent's ability to recover optimal solutions in the Pareto front. It is assumed that we have prior knowledge of the optimal solutions for these benchmarks. Let $B$ be the set of solutions obtained by the agent for various preferences, and let $P$ be the Pareto front set of solutions. The intersection between $B$ and $P$ with a tolerance of $\epsilon$ is defined as $B \cap P := b \in B | \exists\, p \in P : \|b - p\|_1 / \|p\|_1 \le \epsilon$. The precision in this context is defined as $Precision = \frac{|B \cap P|}{|B|}$ and recall is defined as $Recall = \frac{|B \cap P|}{|P|}$. Then CRF1 is computed as $CRF1 = 2\frac{Precision] \times Recall}{Precision + Recall}$. Table 7 summarizes the comparison of PD-MORL with the Envelope algorithm in terms of CRF1, hypervolume, and sparsity metrics on discrete MORL benchmarks. We emphasize that our evaluation process is more extensive than the work proposed by Yang et al. (2019). In their implementation, they report an average of 2000 evaluation episodes where a random preference is sampled uniformly in each episode. In contrast, evaluation of PD-MORL sweeps the entire preference space. The hypervolume and sparsity values for both approaches are obtained through PD-MORL's evaluation process and are discussed in Section 5.1. In this table, we provide the CRF1 values reported in (Yang et al., 2019) for the Envelope algorithm. PD-MORL achieves larger or the same CRF1 values compared to the Envelope algorithm for all benchmarks. Specifically, it achieves up to 12% higher CRF1 for the Fruit Tree Navigation task with a depth of $d = 7$. Additionally, a comparison with (Abels et al., 2019) is given in Table 7 which shows the superiority of both the Envelope algorithm and PD-MORL over the proposed algorithm by Abels et al. (2019) which also learns a unified policy.

Table 7: Comparison of our approach with prior works (Yang et al., 2019; Abels et al., 2019) using discrete MORL benchmarks in terms CRF1, hypervolume, and sparsity metrics.* are the reported values in (Yang et al., 2019).

| | Deep Sea Treasure | | | Fruit Tree Navigation (d=5) | | | Fruit Tree Navigation (d=6) | | | Fruit Tree Navigation (d=7) | | |
| | CRF1 | Hypervolume | Sparsity | CRF1 | Hypervolume | Sparsity | CRF1 | Hypervolume | Sparsity | CRF1 | Hypervolume | Sparsity |
|---|---|---|---|---|---|---|---|---|---|---|---|---|
| Envelope (Yang et al., 2019) | 0.994* | 227.89 | 2.62 | 1 | 6920.58 | N/A | 0.995* | 8427.51 | N/A | 0.819* | 6395.27 | N/A |
| CN+DER (Abels et al., 2019) | 0.989* | – | – | 1 | – | N/A | 0.9258* | – | N/A | 0.6719* | – | N/A |
| Ours | 1 | 241.73 | 1.14 | 1 | 6920.58 | N/A | 1 | 9299.15 | N/A | 0.92 | 11419.58 | N/A |

For the continuous control benchmarks, we provide additional plots on the Pareto front and hypervolume progression. Figure 6(a)-(e) shows the progression of the Pareto front solutions during training. We plot the Pareto front solutions at early stages, mid stages, and late stages of the training process. As the training progresses, the Pareto front expands to a broader and denser curve. This behavior is supported by Figure 7(a)-(e), which shows the progression of the hypervolume for all benchmarks. This figure shows that PD-MORL effectively pushes the hypervolume by discovering new Pareto solutions with the help of efficient and robust exploration. For the MO-HalfCheetah-v2 problem, a broad and dense Pareto front is already achieved at the early stages of the training. Therefore, we do not observe the progression of the Pareto front and hypervolume for this specific benchmark.

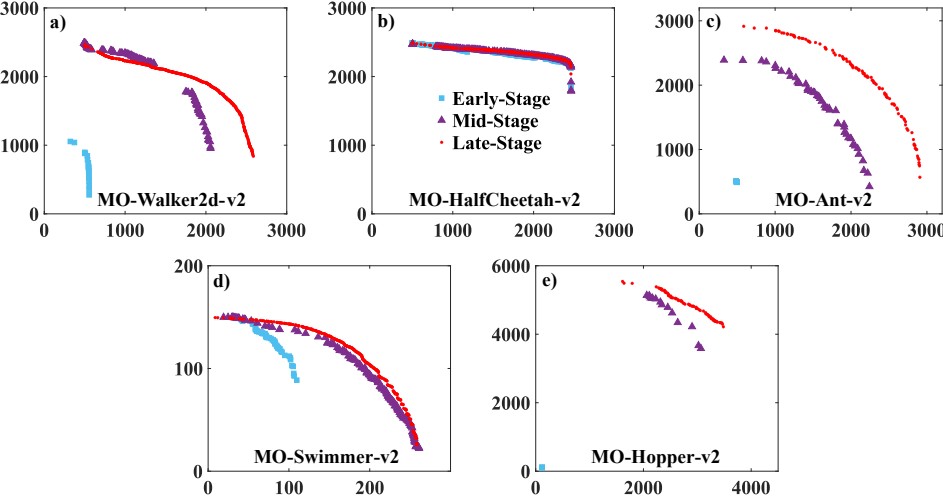

Figure 6: Pareto front progression for (a) MO-Walker-v2, (b) MO-HalfCheetah-v2, (c) MO-Ant-v2, (d) MO-Swimmer-v2, (e) MO-Hopper-v2.

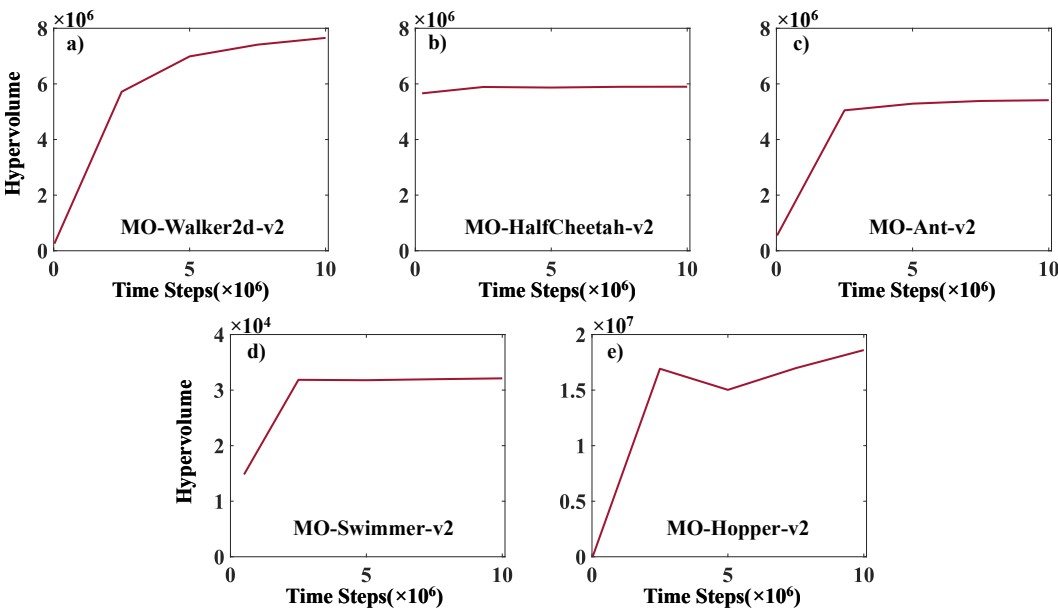

Figure 7: Hypervolume progression for (a) MO-Walker-v2, (b) MO-HalfCheetah-v2, (c) MO-Ant-v2, (d) MO-Swimmer-v2, (e) MO-Hopper-v2.

We also report standard deviations for all benchmarks. As explained in the main manuscript, since META (Chen et al., 2019) and PG-MORL (Xu et al., 2020) report the average of six runs, we also ran each benchmark six times with PD-MORL. Table 8 reports average metrics as well as standard deviations obtained from these six runs. The standard deviations obtained from six runs are not reported in the prior work. This table shows that the individual differences of different runs are mostly three orders of magnitudes less than the average of these runs. This also suggests that PD-MORL achieves a generalizable network independent of starting state of the agent. We also provide more visual results for preferences that captures corner cases for these benchmarks in the supplementary video.

Additionally, we conduct an ablation study to show the effects of the proposed angle term in the loss function and the proposed parallel exploration in our algorithm on continuous control benchmarks. Table 9 reports average metrics and standard deviations obtained from six runs. Since we fit a multi-dimensional interpolator using solutions with key preferences, the interpolator may introduce a bias at the beginning of the training depending on how representative these key solutions are. Hence, this bias may have a negative effect on the training. For tasks with a broad and more convex Pareto front, the probability of this bias increases as we choose key preferences from corner cases.

Table 8: Performance comparison of the proposed approach and state-of-the-art algorithms on the continuous control benchmarks in terms of hypervolume and sparsity metrics. Reference point for hypervolume calculation is set to (0,0) point. HV$^*$: Hypervolume

|  |  | PG-MORL (Xu et al., 2020) | META (Chen et al., 2019) | **PD-MORL (Ours)** |
|---|---|---|---|---|
| **MO-Walker2d-v2** | HV$^*$ | $4.82 \times 10^6$ | $2.10 \times 10^6$ | $\mathbf{5.41 \pm 0.004 \times 10^6}$ |
|  | Sparsity | $0.04 \times 10^4$ | $2.10 \times 10^4$ | $\mathbf{0.03 \pm 0.005 \times 10^4}$ |
| **MO-HalfCheetah-v2** | HV$^*$ | $5.77 \times 10^6$ | $5.18 \times 10^6$ | $\mathbf{5.89 \pm 0.002 \times 10^6}$ |
|  | Sparsity | $\mathbf{0.44 \times 10^3}$ | $2.13 \times 10^3$ | $0.49 \pm 0.041 \times 10^3$ |
| **MO-Ant-v2** | HV$^*$ | $6.35 \times 10^6$ | $2.40 \times 10^4$ | $\mathbf{7.48 \pm 0.019 \times 10^6}$ |
|  | Sparsity | $\mathbf{0.37 \times 10^4}$ | $1.56 \times 10^4$ | $0.78 \pm 0.2 \times 10^4$ |
| **MO-Swimmer-v2** | HV$^*$ | $2.57 \times 10^4$ | $1.23 \times 10^4$ | $\mathbf{3.21 \pm 0.001 \times 10^4}$ |
|  | Sparsity | $9.9$ | $24.4$ | $\mathbf{5.7 \pm 0.7}$ |
| **MO-Hopper-v2** | HV$^*$ | $\mathbf{2.02 \times 10^7}$ | $1.25 \times 10^7$ | $1.88 \pm 0.005 \times 10^7$ |
|  | Sparsity | $0.5 \times 10^4$ | $4.84 \times 10^4$ | $\mathbf{0.3 \pm 0.09 \times 10^4}$ |

Table 9: Ablation study of the proposed approach with and without proposed terms and improvements on continuous control benchmarks in terms of hypervolume and sparsity metrics. Reference point for hypervolume calculation is set to (0,0) point. HV$^*$: Hypervolume

| | | PG-MORL (Xu et al., 2020) | PD-MORL w/o Angle Term | PD-MORL w/o Parallel Exploration | PD-MORL |
|---|---|---|---|---|---|
| **MO-Walker2d-v2** | HV$^*$ | $4.82 \times 10^6$ | $\mathbf{5.51 \pm 0.004 \times 10^6}$ | $2.81 \pm 0.009 \times 10^6$ | $5.41 \pm 0.004 \times 10^6$ |
| | Sparsity | $0.04 \times 10^4$ | $\mathbf{0.02 \pm 0.004 \times 10^4}$ | $0.22 \pm 0.026 \times 10^4$ | $0.03 \pm 0.005 \times 10^4$ |
| **MO-HalfCheetah-v2** | HV$^*$ | $5.77 \times 10^6$ | $5.61 \pm 0.003 \times 10^6$ | $5.84 \pm 0.001 \times 10^6$ | $\mathbf{5.89 \pm 0.002 \times 10^6}$ |
| | Sparsity | $\mathbf{0.44 \times 10^3}$ | $49.73 \pm 3.917 \times 10^3$ | $1.218 \pm 0.172 \times 10^3$ | $0.49 \pm 0.041 \times 10^3$ |
| **MO-Ant-v2** | HV$^*$ | $6.35 \times 10^6$ | $\mathbf{9.12 \pm 0.017 \times 10^6}$ | $4.98 \pm 0.237 \times 10^6$ | $7.48 \pm 0.019 \times 10^6$ |
| | Sparsity | $\mathbf{0.37 \times 10^4}$ | $\mathbf{0.58 \pm 0.09 \times 10^4}$ | $0.91 \pm 0.25 \times 10^4$ | $0.78 \pm 0.2 \times 10^4$ |
| **MO-Swimmer-v2** | HV$^*$ | $2.57 \times 10^4$ | $2.78 \pm 0.001 \times 10^4$ | $3.21 \pm 0.001 \times 10^4$ | $\mathbf{3.21 \pm 0.001 \times 10^4}$ |
| | Sparsity | $9.9$ | $6.3 \pm 0.9$ | $9.4 \pm 1.1$ | $\mathbf{5.7 \pm 0.7}$ |
| **MO-Hopper-v2** | HV$^*$ | $\mathbf{2.02 \times 10^7}$ | $1.92 \pm 0.023 \times 10^7$ | $0.63 \pm 0.005 \times 10^7$ | $1.88 \pm 0.005 \times 10^7$ |
| | Sparsity | $0.5 \times 10^4$ | $6.1 \pm 2.30 \times 10^4$ | $20.53 \pm 33.20 \times 10^4$ | $\mathbf{0.3 \pm 0.09 \times 10^4}$ |

Therefore, by removing the angle term in the loss function and thus, removing the bias, the performance of our algorithm slightly degrades due to the convex and broad nature of the Pareto front in MO-Walker2d-v2 and MO-Ant-v2 problems. However, removing the angle term has a negative effect for MO-HalfCheetah-v2 and MO-Hopper-v2, where the Pareto front is dense. Figure 8 shows the obtained Pareto front with and without the directional angle term in the loss function. The solutions obtained without the directional angle term are sparse. Specifically, for MO-HalfCheetah-v2, Table 9 shows that both the hypervolume and the sparsity metrics are negatively affected. For MO-Hopper-v2, although the hypervolume metric seems to be increased, the change in the sparsity metric suggests that the algorithm cannot find a dense Pareto front. This, in fact, supports that the hypervolume alone is insufficient to assess the quality of the Pareto front. We further investigate the effects of using parallel exploration. To this end, we collect transitions using a single preference space instead of dividing them into sub-spaces. Parallel exploration guarantees that obtained transitions are not biased towards a preference sub-space. Hence, it increases the sample efficiency of the algorithm. Table 9 shows that removing the parallel exploration significantly reduces the performance of the algorithm. For all tasks, the hypervolume metric decreases, and the sparsity metric increases. We also observe that both PG-MORL and PD-MORL with proposed novel aspects are superior to PD-MORL without the novel aspects. This observation is crucial since the prior work uses PPO, whereas PD-MORL uses TD3 in their algorithm pipeline. Using this observation, we conclude that the superiority of PD-MORL is due to the proposed novel aspects.

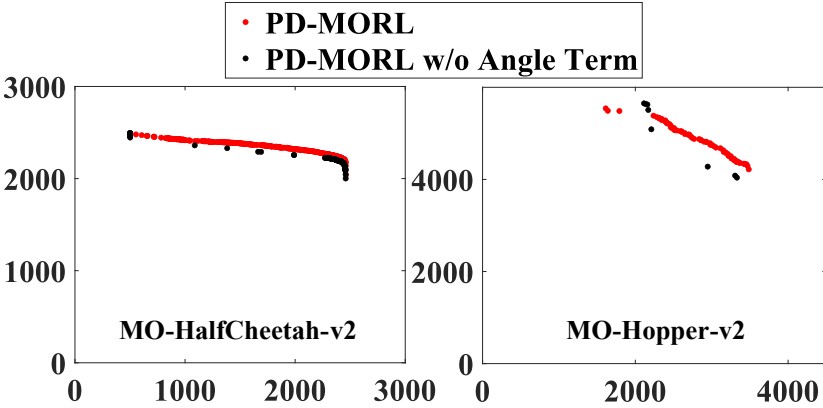

Figure 8: Pareto front with and without the directional angle term in the loss function for MO-HalfCheetah-v2 and MO-Hopper-v2 problems. In the latter case, the solutions are sparse and represented by few points.

