# OpenReview forum: "PD-MORL: Preference-Driven Multi-Objective Reinforcement Learning Algorithm"
_ICLR.cc/2023/Conference — ICLR 2023 poster_

### Official Review · Reviewer_sFJy · 2022-10-24

**Confidence:** 3
**Correctness:** 4
**Technical Novelty And Significance:** 3
**Empirical Novelty And Significance:** 3
**Recommendation:** 8

**Clarity, Quality, Novelty And Reproducibility:**

Besides the main weakness described above, the work is clearly presented and provides a novel contribution.

Small comments and typos:
- Section 2: the last sentence is repeated almost identically twice
- Section 3, definition 1: "For the same P be a Pareto front" - what does for the same P means here? (it seems there is a grammar issue in this sentence)
- Section 4, paragraph 1: optimalioty (typo)
- Section 4.1, below eq. (3): identitity (typo)
- Section 4.1, theorems 3 and 4: "Let (Q,d) is" -> "Let (Q,d) be"
- Section 4.1, below theorem 3: "evaluation and optimality operators are contraction" -> "contractions"
- Section 4.2, first paragraph: "to obtain a single parameterized function represents" -> representing? that represents?
- Section 5: "Let the first objective is the treasure value and the second objective is" -> be and be

**Strength And Weaknesses:**

The main strengths of this paper:
- Novel approach to solve MORL with a single network
- Extensive experiments both in discrete and continuous state-action spaces
- Theoretical support of optimality

The main weakness of this paper:
- Some details are lacking in Section 4.2, specifically concerning the division of the preference space, the preference alignment using normalization and more importantly the extension of the algorithm to continuous action space. While it is true that these details are provided in the supplementary material, it would be helpful to have some hints in the main text. Space could be saved by condensing the introduction and related work section, which are rather detailed, thus leaving more space to describe the main novel components of this work.

**Summary Of The Paper:**

This work proposes an algorithm for solving multi-objective reinforcement learning (MORL) problems with a single network, termed preference-driven MORL (PD-MORL). The main insight to do so (instead of computing a policy for each potential preference)

**Summary Of The Review:**

This work provides a strong contribution, back up with theoretical results as well as empirical experiments.

---

> ### Author Response · Authors · 2022-11-16
> **Response to Reviewer sFJy**
>
> We thank the reviewer for their time, effort and supportive review.
>
> **C1:** Some details are lacking in Section 4.2, specifically concerning the division of the preference space, the preference alignment using normalization and more importantly the extension of the algorithm to continuous action space. While it is true that these details are provided in the supplementary material, it would be helpful to have some hints in the main text. Space could be saved by condensing the introduction and related work section, which are rather detailed, thus leaving more space to describe the main novel components of this work.
>
> **A1:** As the reviewer suggested, we have revised Section 4.2 to let the readers better understand the details of our algorithm in the main manuscript. Specifically, we have added
>
> * A detailed explanation of the benefit of the cosine similarity term in the optimality operator in Section 4.2 after Equation 6.
>
> * A hint on why HER is preferred in the second paragraph of Section 4.2.
>
> * A more detailed explanation of the preference alignment procedure in the "Preference alignment" paragraph of Section 4.2.
>
> * A more detailed explanation of MO-TD3-HER extension for continuous action space in the "Extension to continuous action space" paragraph of Section 4.2.
>
> The modifications/additions to the original manuscripts are highlighted with blue in the revised manuscript.
>
> **C2:** Small comments and typos:
>
> **A2:** We also fixed the typos in the revised manuscript.

---

> > ### Comment · Reviewer_sFJy · 2022-11-27
> > **Response to the authors**
> >
> > Dear Authors,
> >
> > Thank you for your response as well as the updates performed in the manuscript.

---

> > > ### Author Response · Authors · 2022-11-29
> > > **Response to Reviewer sFJy's Feedback on Our Initial Response**
> > >
> > > Dear Reviewer sFJy,
> > >
> > > Thank you for your feedback! Your suggestions and comments definitely help us to improve our work. We will further update/improve our work based on the discussion with other reviewers.

---

### Official Review · Reviewer_scaX · 2022-10-25

**Confidence:** 4
**Correctness:** 3
**Technical Novelty And Significance:** 4
**Empirical Novelty And Significance:** 3
**Recommendation:** 8

**Clarity, Quality, Novelty And Reproducibility:**

The paper is mostly well written and clear, aside from a few monolithic paragraphs that should be better split up and organized (particularly in sections 2 & 4). The quality of the experiments and results is good, and a skim of the appendix makes me think it has solid coverage of the details. The main challenge here is demonstrating a significant and consistent improvement over the baselines, which was discussed as a weakness earlier. The main strength of the paper is probably the novelty and significance of the approach -- the paper makes a compelling case for why this type of approach should be used going forward (order of magnitude fewer parameters, single policy to cover any/all preferences), which I think makes the paper interesting and relevant to the community even though the empirical results are more marginal. The reproducibility statement reassures that the results will be reproducible, and the usage of standard benchmarks is well appreciated.

**Strength And Weaknesses:**

The paper excels at explaining the problem, related works, algorithm, and results. The clarity of most sections is very good and the design of the algorithm, choice of experiments, metrics, figures, etc are all well justified and effective. It also very effectively highlights the differences of PD-MORL from previous works, and justifies why these differences make this approach a promising direction for multi-objective reinforcement learning. Figures 4 and 5 are also excellent at visualizing the algorithm and key ingredients to its success.

There are a few weaknesses in the paper too, however. First, MORL benchmarks used are not very well explained and the visualizations for the tasks and policies are not very clear. I have minimal understanding of what the challenges in each benchmark are and why baselines failed in these tasks -- this paper does not need an exhaustive description of these benchmarks but could certainly explain them and the learned policies better through some textual revisions and more figures (e.g., fig 3 is not very well explained/useful in its current state). Next, the experimental results are not sufficiently explored. For example, the paper does not address why the contributed PD-MORL performs significantly worse than PG-MORL in the hopper experiment, or why sparsity is sometimes worse than PG-MORL. Including these results is essential, but so is some analysis insight into the weaknesses of PD-MORL relative to PG-MORL in these test cases, and perhaps also some ideas on how these issues in PD-MORL might be overcome. It also seems worth including some discussion on the limitations of these metrics. On a related note, the results in Table 1 should have error values representing. Thirdly, the usage of the cosine similarity in eq. 5 is not well enough explained -- the inner product of omega and Q should already reflect the cosine similarity of omega and Q(s,a, omega), so why is the scalar Sc(omega, Q) also multiplied by the inner product of omega and Q? I also don't fully understand the preference alignment procedure, including how the normalized solutions in Fig 2b are formed and what purpose there is to interpolating them, e.g. rather than interpolating in a different space. Finally, the related works and experiments sections are rather monolithic and would benefit greatly from splitting the large paragraphs into individual ideas or techniques, possibly using other methods (e.g. figures/tables and subsections) to better organize and discuss the ideas in them.

A few more small notes: the vector nature of Q is not well explained (why is the dimensionality of Q(s, a, w) equal to L and what does it represent before/after taking the inner product with omega?), the walker results in fig 5 need y-axis labels, the paper should justify why fewer parameters is a big deal, the limitations/future work discussion isn't very interesting/compelling and should be more specific and try to offer some deeper insights. Also, the use of L2 distance in sparsity is possibly worth discussing if there's space -- it seems like a rather arbitrary choice of metric so I am curious if there is a reason for it.



**Summary Of The Paper:**

The paper presents PD-MORL as a fascinating new method to learn a single policy that can integrate task preferences and cover more of the Pareto front than any baseline algorithm. This differentiates it from previous methods that must learn a new policy for any particular preference. It demonstrates marginally better results in existing multi-objective baseline problems, with excellent analysis to explain the improvement.

**Summary Of The Review:**

This is a good paper that takes a clever, novel, and not overly complex approach to multi-objective reinforcement learning. While the empirical results aren't outstanding and the description of the algorithm itself is a little unclear, the rest of the analysis is solid and overall everything seems sufficiently suitable and solid for this conference. I think this is sure to inspire good follow-up work, further justifying why it should be shared with the wider community.

Update: The author responses have addressed all of my earlier concerns and the few new concerns are very minor. I think the changes address most of my fellow reviewers concerns and improved the paper overall, so I have decided to maintain my score of 8.

---

> ### Author Response · Authors · 2022-11-16
> **Response to Reviewer scaX**
>
> We thank the reviewer for their time, effort, and supportive review. We want to emphasize that all the points the reviewer pointed out are towards improving the quality of this work, and we appreciate it. The modifications/additions to the original manuscripts are highlighted in blue in the revised manuscript.
>
> **C1:** First, MORL benchmarks used are not very well explained and the visualizations for the tasks and policies are not very clear. I have minimal understanding of what the challenges in each benchmark are and why baselines failed in these tasks -- this paper does not need an exhaustive description of these benchmarks but could certainly explain them and the learned policies better through some textual revisions and more figures (e.g., fig 3 is not very well explained/useful in its current state).
>
> **A1:** We provide additional details for benchmarks in Sections 5.1 and 5.2. We also refer to Appendix B.2. at the beginning of Section 5 to lead the readers to detailed descriptions of the benchmarks used in this work. As the reviewer suggested, we modified the second paragraph of Section 5.1, where we explain and refer to Figure 3 as follows:
>
> "Figure 3 illustrates the policies achieved by PD-MORL for three preference vectors capture corner cases. Although our policy is not specific to any preference, the submarine achieves optimal treasure values for the given preference. For example, it finds the best trade-off within the shortest time (one step) when we only value the time penalty ($w=$\{0,1\}). As the importance of the treasure value increases ($w=$\{0.5,0.5\}, $w=$\{1,0\}), it spends optimal amount of time (8 steps and 19 steps, respectively) to find a deeper treasure."
>
> **C2:** Next, the experimental results are not sufficiently explored. For example, the paper does not address why the contributed PD-MORL performs significantly worse than PG-MORL in the hopper experiment, or why sparsity is sometimes worse than PG-MORL. Including these results is essential, but so is some analysis insight into the weaknesses of PD-MORL relative to PG-MORL in these test cases, and perhaps also some ideas on how these issues in PD-MORL might be overcome. It also seems worth including some discussion on the limitations of these metrics.
>
> **A2:** We agree with the reviewer that a more detailed discussion of the experimental results is missing. We have revised the third paragraph in Section 5.2 to address the reviewer's concerns and added a discussion as follows:
>
> "We also note that PG-MORL achieves better sparsity metric for MO-HalfCheetah-v2 and MO-Ant-v2 environments. However, PG-MORL also achieves smaller hypervolume compared to PD-MORL. This suggests that to determine the quality of a Pareto front, these metrics are rather limited and should be further investigated by the existing literature. Therefore, we also plot Pareto front plots for all algorithms in Figure 5(a)-(e) to have a better understanding of the quality of the Pareto front."
>
> "PD-MORL mostly relies on the interpolation procedure before the actual training. The key solutions obtained during this period determine the initial convergence of the algorithm. However, for MO-Hopper-v2, the obtained key solutions for preference vectors \{1,0\}, \{0.5,0.5\}, \{0,1\} are \{1778,4971\}, \{1227, 1310\}, \{3533, 3165\} respectively. These key solutions are not a good representative of the Pareto front shown in Figure 5(e). To have representative key solutions is a limitation to PD-MORL; however, it can be solved with hyperparameter tuning."
>
> **C3:** On a related note, the results in Table 1 should have error values representing.
>
> **A3:** Since these benchmarks are relatively simpler than the continuous control tasks, we specifically reported the mean values over different seeds in Table 1 and Table 7. The results are extremely similar using different seeds.

---

> > ### Author Response · Authors · 2022-11-16
> > **Response to Reviewer scaX - Cont.**
> >
> > **C4:** Thirdly, the usage of the cosine similarity in eq. 5 is not well enough explained -- the inner product of omega and Q should already reflect the cosine similarity of omega and Q(s,a, omega), so why is the scalar Sc(omega, Q) also multiplied by the inner product of omega and Q?
> >
> > **A4:** We thank the reviewer for pointing out this confusion of redundancy.
> >
> > Without the cosine similarity term, the supremum operator yields the action that only maximizes $\boldsymbol{\omega}^T\mathbf{Q}(s^\prime,a^{\prime},\boldsymbol{\omega})$. This may pull the target Q-values in the wrong direction, especially when the scales of the objectives are in different orders of magnitude. For instance, let us assume a multi-objective problem with percent energy efficiency ($\in [0, 1]$) being the first objective and the latency ($\in [1, 10]$) in milliseconds being the second objective. Let us also assume that the preference vector ($\boldsymbol{\omega} =$ \{$0.9, 0.1$\}) favors the energy efficiency objective, and there are two separate actions ($a_1, a_2$) that yields Q-values of $Q_1 =$\{$0.9, ~1$\}, $Q_2 = $\{$0.1, ~10$\} respectively. The supremum operator chooses action, $a_2$ since $\boldsymbol{\omega}^T Q_2=1.09$ is higher than $\boldsymbol{\omega}^T Q_1=0.91$. Since the scales of these two objectives are in different orders of magnitudes, $\sup_{\substack{a^{\prime}\in A}}(\boldsymbol{\omega}^T\mathbf{Q}s^\prime,a^{\prime},\boldsymbol{\omega}))$ always chooses the action that favors the latency objective. This behavior negatively affects the training since the target Q-values are pulled in the wrong direction. On the contrary, our novel cosine similarity term enables the optimality operator to choose actions that align the preferences with the Q-values and maximize the target value at the same time. The supremum operator now chooses action, $a_1$ since $S_c\cdot\boldsymbol{\omega}^T Q_1=0.75\times0.91 = 0.68$ is higher than $S_c\cdot\boldsymbol{\omega}^T Q_2=0.12\times1.09=0.13$. With this addition, the algorithm disregards the large  $\boldsymbol{\omega}^T\mathbf{Q}(s^\prime,a^{\prime},\boldsymbol{\omega})$ where the preference vector and the Q-values are not aligned since the $S_c(\boldsymbol{\omega},\mathbf{Q}(s^\prime,a^{\prime},\boldsymbol{\omega}))$ takes small values.
> >
> > We added this detailed explanation of the benefit of using the cosine similarity term in Section 4.2 after Equation 6.
> > The modifications/additions to the original manuscripts are highlighted in blue in the revised manuscript.
> >
> > **C5:** I also don't fully understand the preference alignment procedure, including how the normalized solutions in Fig 2b are formed and what purpose there is to interpolating them, e.g. rather than interpolating in a different space.
> >
> > **A5:** Most of the missing details were provided in the Appendix. Based on the reviewers' suggestions, we have revised the "Preference alignment" paragraph in Section 4.2 to let the readers better understand the details of our algorithm in the main manuscript.
> >
> > **C6:** Finally, the related works and experiments sections are rather monolithic and would benefit greatly from splitting the large paragraphs into individual ideas or techniques, possibly using other methods (e.g. figures/tables and subsections) to better organize and discuss the ideas in them.
> >
> > **A6:** Since we added new details in the revised manuscript, we condensed some of the sections to eliminate the monolithic nature of these sections.
> >
> > **C7:** The vector nature of Q is not well explained (why is the dimensionality of Q(s, a, w) equal to L and what does it represent before/after taking the inner product with omega?).
> >
> > **A7:** The vector of nature Q is first introduced in the Background section, and L is an arbitrary choice of dimension. Similarly, in the same section, we explained the scalar nature of the inner product. These details are also provided in Appendix A.
> >
> > **C8:** The walker results in fig 5 need y-axis labels
> >
> > **A8:** We have added x-axis and y-axis labels to Figure 5.

---

> > > ### Author Response · Authors · 2022-11-16
> > > **Response to Reviewer scaX - Cont.**
> > >
> > > **C9:** The paper should justify why fewer parameters is a big deal, the limitations/future work discussion isn't very interesting/compelling and should be more specific and try to offer some deeper insights.
> > >
> > > **A9:** In the MORL community, the multi-objective nature of the algorithm is generally motivated by the fact that real-world tasks have multiple, possibly conflicting objectives. However, in order for them to be practical, MORL approaches should be deployable in real-world scenarios. Hence, they should be memory-efficient and energy-efficient. Using this insight, we have revised the third paragraph in Section 5.2 and added the following sentence:
> > >
> > > "This may enable the deployment of PD-MORL in a real-life scenario where there are dynamic changes in the design constraints and goals.
> > >
> > >
> > > **C10:** Also, the use of L2 distance in sparsity is possibly worth discussing if there's space -- it seems like a rather arbitrary choice of metric so I am curious if there is a reason for it.
> > >
> > > **A10:** We thank the reviewer for discussing the use of L2 distance in sparsity. We agree with the reviewer that L2 distance has its own limitations, especially when the objective space has large dimensions. However, we employ this metric to have a fair comparison with the existing literature.
> > >
> > > We have revised the third paragraph in Section 5.2 and state that "to determine the quality of a Pareto front, these metrics are rather limited and should be further investigated by the existing literature." in the revised manuscript.

---

> > > > ### Comment · Reviewer_scaX · 2022-11-28
> > > > **Thank you for detailed response**
> > > >
> > > > Thank you for the detailed response to each of my comments. You have addressed all of the concerns I raised (your explanation about the cosine similarity was particularly enlightening) and I appreciate the changes made to the paper. I have also noted the issues raised by my fellow reviewers but your responses seem to do well at addressing those as well. I think this paper is now a solid 8 so I will keep recommending acceptance.
> > > >
> > > > Based on the updates, I have one new, albeit minor, concern: are you sure the solution in Fig 3(b) is actually optimal? Based on the preference vector, isn't it worth taking 4 more steps (12 total) to reach the 19.6 reward. The other two examples make sense to me.
> > > >
> > > > I also have a few editorial notes. Section 4.2 feels quite verbose overall. I have one idea to make it a bit more concise: while I think the example you gave on energy efficiency versus latency was helpful for understanding the cosine similarity term, I'd be inclined to either make the example a bit more concise or move it to Appendix B (including a reference to it in the main text).
> > > >
> > > > Also, the paragraph starting with "For each episode during training" is a little confusing to read. I'd tweak and reorder the sentences as something like:
> > > > 1. For each episode during training...
> > > > 2. However, this approach may create a bias towards...
> > > > 3. To see this, suppose the randomly sampled preference vector favors energy efficiency...
> > > > 4. Hence,...
> > > > 5. To overcome this issue, we employ...
> > > >
> > > > Lastly, I'd encourage doing a final pass for grammar+clarity in the final copy. It's pretty good overall but there are a few issues, for example, "should be further investigated by the existing literature" doesn't make sense and should be replaced with "should be further investigated by future works".

---

> > > > > ### Author Response · Authors · 2022-11-29
> > > > > **Response to Reviewer scaX's Feedback on Our Initial Response**
> > > > >
> > > > > Thank you for your feedback! Once again, we want to point out that your suggestions and comments help us to improve the quality of our work, and we appreciate it.
> > > > >
> > > > > **C1:** Based on the updates, I have one new, albeit minor, concern: are you sure the solution in Fig 3(b) is actually optimal? Based on the preference vector, isn't it worth taking 4 more steps (12 total) to reach the 19.6 reward. The other two examples make sense to me.
> > > > >
> > > > > **A1:** We thank the reviewer for pointing out this error. We realized that there was an error in the figure. The distance between the treasure values 16.1 and 19.6 should be 3 cells, not 2. We have fixed it but cannot submit a revised version during this period. We will pursue this in the final copy if the paper gets accepted.
> > > > >
> > > > > **C2:** I also have a few editorial notes. Section 4.2 feels quite verbose overall. I have one idea to make it a bit more concise: while I think the example you gave on energy efficiency versus latency was helpful for understanding the cosine similarity term, I'd be inclined to either make the example a bit more concise or move it to Appendix B (including a reference to it in the main text).
> > > > >
> > > > > **A2:** We agree with the reviewer that this section is verbose. Since the cosine similarity term is one of the main novelties of our work and the effectiveness of this term is a common concern among reviewers, we want to emphasize and explain it in the main manuscript rather than in Appendix. We thank the reviewer for the input and will consider it for the final copy based on the discussion during this period.
> > > > >
> > > > > **C3:** Also, the paragraph starting with "For each episode during training" is a little confusing to read. I'd tweak and reorder the sentences as something like:
> > > > >
> > > > > * For each episode during training...
> > > > > * However, this approach may create a bias towards...
> > > > > * To see this, suppose the randomly sampled preference vector favors energy efficiency...
> > > > > * Hence,...
> > > > > * To overcome this issue, we employ...
> > > > >
> > > > > **A3:** We thank the reviewer for their suggestion. The suggested structure reads better and will be used in the final copy.
> > > > >
> > > > > **C4:** Lastly, I'd encourage doing a final pass for grammar+clarity in the final copy. It's pretty good overall but there are a few issues, for example, "should be further investigated by the existing literature" doesn't make sense and should be replaced with "should be further investigated by future works".
> > > > >
> > > > > **A4:** We thank the reviewer for this feedback. We will do a detailed final pass and correct any grammar/clarity errors.

---

### Official Review · Reviewer_Mx6C · 2022-10-27

**Confidence:** 3
**Correctness:** 3
**Technical Novelty And Significance:** 3
**Empirical Novelty And Significance:** 3
**Recommendation:** 5

**Clarity, Quality, Novelty And Reproducibility:**

The paper is clearly written. The major novelty of the paper is the cosine similarity term in the optimality operator. This work has good reproducibility as the authors provide source code in the submission.

**Strength And Weaknesses:**

- Multi-objective reinforcement learning is an important task, and learning a single network for multiple preferences is an efficient approach to it.
- Theoretical analysis is provided. The authors proved (in the appendix) that their optimality operator is a contraction and the optimal value function is a fixed point for the optimality operator.
- The effectiveness of the cosine similarity term in equation 5. In equation 5, we have cos(w, Q)•w^T•Q in the supremum operator. However, w^T•Q=cos(w, Q)•|w|•|Q|, which already includes the cosine similarity term. Is there a specific reason we need an extra cosine similarity term? A good way to verify its effectiveness is to perform an ablation study. Note that simply removing the cosine similarity term is not equivalent to Envelope as the supremum in the Envelope algorithm takes over different preferences.

----Rebuttal----
Dear authors,

Thank you for your detailed response. Unfortunately, I'm afraid I disagree with your explanation of the cosine similarity term. In your explanation, if we simply change the range of latency to [1, 100] and set Q2 to {0.1, 100}, even with the cosine similarity term, we still will prefer action 2 (you can verify as 0.68 < 1.12). Therefore, if we have to think of w as a preference vector, the cosine similarity term is an inappropriate approach to fix the issue you mentioned. Instead, we can rescale and normalize the magnitude of all the objectives, which easily fixes the issue you mentioned.

As I can't agree with your explanation of the cosine similarity term, I have to lower my rating of the paper.

**Summary Of The Paper:**

This paper proposes PD-MORL, which extends Envelope (Yang et al., 2019). The major difference is illustrated in Equation 5 in the paper: an extra cosine similarity term is added, and the supremum is only taken over actions, not preferences (compared to Envelope). The authors evaluated their method with Envelope on Deep Sea Treasure and Fruit Tree Navigation, and with PG-MORL and META on some MuJoCo continuous control tasks.

**Summary Of The Review:**

This paper proposes a new method of MORL based on Envelope, where they include a cosine similarity term in their optimality operator. They provided a theoretical analysis of their method, as well as performance evaluations with several baselines on several different tasks. My major concern in this work is the effectiveness of the cosine similarity term, and I believe an ablation study can demonstrate it.

---

> ### Author Response · Authors · 2022-11-16
> **Response to Reviewer Mx6C**
>
> We thank the reviewer for their time, effort and supportive review.
>
> **C1:** Effectiveness of cosine similarity term.
>
> **A1:** We thank the reviewer for pointing out this confusion of redundancy.
>
> Without the cosine similarity term, the supremum operator yields the action that only maximizes $\boldsymbol{\omega}^T\mathbf{Q}(s^\prime,a^{\prime},\boldsymbol{\omega})$. This may pull the target Q-values in the wrong direction, especially when the scales of the objectives are in different orders of magnitude. For instance, let us assume a multi-objective problem with percent energy efficiency ($\in [0, 1]$) being the first objective and the latency ($\in [1, 10]$) in milliseconds being the second objective. Let us also assume that the preference vector ($\boldsymbol{\omega} =$ \{$0.9, 0.1$\}) favors the energy efficiency objective and there are two separate actions ($a_1, a_2$) that yields Q-values of $Q_1 =$\{$0.9, ~1$\}, $Q_2 = $\{$0.1, ~10$\} respectively. The supremum operator chooses action, $a_2$ since $\boldsymbol{\omega}^T Q_2=1.09$ is higher than $\boldsymbol{\omega}^T Q_1=0.91$. Since the scales of these two objectives are in different orders of magnitudes, $\sup_{\substack{a^{\prime}\in A}}(\boldsymbol{\omega}^T\mathbf{Q}s^\prime,a^{\prime},\boldsymbol{\omega}))$ always chooses the action that favors the latency objective. This behavior negatively affects the training since the target Q-values are pulled in the wrong direction. On the contrary, our novel cosine similarity term enables the optimality operator to choose actions that align the preferences with the Q-values and maximize the target value at the same time. The supremum operator now chooses action, $a_1$ since $S_c\cdot\boldsymbol{\omega}^T Q_1=0.75\times0.91 = 0.68$ is higher than $S_c\cdot\boldsymbol{\omega}^T Q_2=0.12\times1.09=0.13$. With this addition, the algorithm disregards the large  $\boldsymbol{\omega}^T\mathbf{Q}(s^\prime,a^{\prime},\boldsymbol{\omega})$ where the preference vector and the Q-values are not aligned since the $S_c(\boldsymbol{\omega},\mathbf{Q}(s^\prime,a^{\prime},\boldsymbol{\omega}))$ takes small values.
>
> We have added this detailed explanation of the benefit of using cosine similarity term in Section 4.2 after Equation 6.
> The modifications/additions to the original manuscripts are highlighted with blue in the revised manuscript.
>
> We also note that, we already had an ablation study where we remove the cosine similarity term from the loss function in Appendix B.4 for continuous control tasks. In this section, we have discussed the results on this ablation study.

---

> > ### Author Response · Authors · 2022-12-05
> > **Response to Reviewer Mx6C - Looking forward to hearing your feedback**
> >
> > Dear Reviewer Mx6C,
> >
> > Thanks again for your time, effort, and your supportive comments to improve our work! We have addressed your initial concerns. We are looking forward to hearing your feedback and will be happy to answer/discuss any further questions/concerns you may have.
> >
> > Best regards,

---

### Official Review · Reviewer_QTgN · 2022-11-02

**Confidence:** 3
**Correctness:** 3
**Technical Novelty And Significance:** 2
**Empirical Novelty And Significance:** 2
**Recommendation:** 3

**Clarity, Quality, Novelty And Reproducibility:**

Most of my clarity comments are in the above section.

I have one question about using the cosine similarity in the preference-driven optimality operator. The cosine similarity is closely related to the inner product of the preference vector and the Q value vector, which is also in the operator itself. So is it really important to have the $\omega^t Q(s’, a’, \omega)$ term in the update rule, or is the cosine similarity itself sufficient?

In theorem 4, the authors should define what Q* is as it is not clear in the multi-objective setting.


**Strength And Weaknesses:**

Strength:
The idea to align the preference vector better with the Q function is interesting and relevant to the multi-objective RL community.

The writing is mostly clear and easy to follow.

Weakness:
There are important baselines missing in the empirical evaluations. The envelope Q-learning algorithm by Yang et al. is only compared in DST and FTN but missing from MuJoCo environments.

Another important baseline is the conditional network with diverse experience replay algorithm in Abels et al. (Dynamic Weights in Multi-Objective Deep Reinforcement Learning). I believe adding both baselines would make the comparisons more informative.

There seems to be a mismatch between the practical version of the algorithm and the theoretical version. Algorithm 1 uses the interpolated preference rather than the raw preference. Can the authors comment on this choice and how the theoretical result extends to this different update rule?

There are some details missing in the description of the algorithm. The preference interpolation is an important part of the algorithm but its description is lacking. I suggest having a more detailed exposition on this, including how the key preference points are selected, how the normalization procedure is done, and a mathematical expression of the interpolation function.


**Summary Of The Paper:**

In this paper, the authors propose a new multi-objective RL algorithm named Preference-Driven MORL (PD-MORL). PD-MORL uses the preference to align the Q function updates. Theoretical analysis shows that the new preference-driven optimality operator ensures convergence to the optimal Q function. Experiments on classical benchmark environments and multi-objective variants of MuJoCo environments are provided.

**Summary Of The Review:**

Because of the important baseline results missing, I am not convinced that the paper in its current form is ready to be published.

---

> ### Author Response · Authors · 2022-11-16
> **Response to Reviewer QTgN**
>
> We thank the reviewer for their time and effort and address their concerns.
>
> **C1:** The envelope Q-learning algorithm by Yang et al. is only compared in DST and FTN but missing from MuJoCo environments.
>
> **A1:** As the reviewer stated, we showed that PD-MORL is superior to the Envelope approach in simpler environments with discrete action spaces, DST and FTN. The authors who proposed the Envelope algorithm have not evaluated it using the MuJoCo environments. All the environments used in their paper have discrete action spaces. Therefore, the implementation for continuous action spaces is missing. We also believe that the Envelope approach is not scalable to benchmarks with continuous action spaces such as MuJoCo environments. Since PD-MORL outperforms the envelope algorithm even in simpler environments, its benefits on more complex problems with continuous action spaces are expected to be higher.
>
> Our original justification in Section 5.2 may not have been clear. We clarified our choice as follows:
>
> "We compare the performance of PD-MORL against two state-of-the-art approaches (Xu et al., 2020;Chen et al., 2019) that use these continuous control benchmarks. We emphasize that these approaches must learn a different policy network for every solution on the Pareto front. In contrast, PD-MORL learns a single universal network that covers the entire preference space. The Envelope algorithm (Yang et al., 2019) is not included in these comparisons since it was not evaluated with continuous action spaces, and PD-MORL already outperforms it on simpler problems."
>
> **C2:** Another important baseline is the conditional network with diverse experience replay algorithm in Abels et al. (Dynamic Weights in Multi-Objective Deep Reinforcement Learning).
>
> **A2:** We thank the reviewer for pointing out the work by Abels et al. 2019. We are aware of this work and have already mentioned it in the main manuscript in Sections 2 and 5. In their work, Yang et al. 2019, shows the superiority of the Envelope algorithm over the CN+DER approach provided by Abels et al.. Therefore, we only included the CN+DER approach as a comparison point in Appendix B.4 Table 7 regarding the CRF1 metric.
>
> Additionally, Abels et al. evaluate their work only on image-based simple problems with discrete action spaces (Deep Sea Treasure and Minecart). This and the fact that their network architecture is based on Q-network suggest that their approach cannot be extended to problems with continuous action spaces such as MuJoCo environments.
>
> **C3:** There seems to be a mismatch between the practical version of the algorithm and the theoretical version. Algorithm 1 uses the interpolated preference rather than the raw preference. Can the authors comment on this choice and how the theoretical result extends to this different update rule?
>
> **A3:** We thank the reviewer for pointing out the potential confusion between the practice and theory regarding the use of projected preference in the optimality operator.
> The proof of Theorem 3 in Appendix A.2 assumes that an action $a^{\prime}$ is the action that maximizes $S_c(\boldsymbol{\omega},\mathbf{Q}s^\prime,a^{\prime},\boldsymbol{\omega}))\cdot(\boldsymbol{\omega}^T\mathbf{Q}s^\prime,a^{\prime},\boldsymbol{\omega}))$. This assumption is valid for projected preference $\boldsymbol{\omega}_p$. Therefore, the theoretical results extend this practical modification. To avoid any confusion, we have also added the following sentence to the "Preference alignment" paragraph in Section 4.2:
>
> "This practical modification extends to theoretical results as the proof for Theorem 3 is also valid for $\boldsymbol{\omega}_p$."
>
> **C4:** There are some details missing in the description of the algorithm. The preference interpolation is an important part of the algorithm but its description is lacking. I suggest having a more detailed exposition on this, including how the key preference points are selected, how the normalization procedure is done, and a mathematical expression of the interpolation function.
>
> **A4:** Based on the reviewers' suggestions, we have revised the "Preference alignment" paragraph in Section 4.2 to let the readers better understand the details of our algorithm in the main manuscript. These details were described before only in Appendix B.1.3. We further provide an additional reference for the radial basis function interpolator used in our work for interested readers in Appendix B.1.3. The modifications/additions to the original manuscripts are highlighted in blue in the revised manuscript.

---

> > ### Author Response · Authors · 2022-11-16
> > **Response to Reviewer QTgN - Cont.**
> >
> > **C5:** I have one question about using the cosine similarity in the preference-driven optimality operator. The cosine similarity is closely related to the inner product of the preference vector and the Q value vector, which is also in the operator itself. So is it really important to have the $\omega^TQ(s',a',\omega)$ term in the update rule, or is the cosine similarity itself sufficient?
> >
> > **A5:** We thank the reviewer for pointing out this confusion of redundancy.
> >
> > Without the cosine similarity term, the supremum operator yields the action that only maximizes $\boldsymbol{\omega}^T\mathbf{Q}(s^\prime,a^{\prime},\boldsymbol{\omega})$. This may pull the target Q-values in the wrong direction, especially when the scales of the objectives are in different orders of magnitude. For instance, let us assume a multi-objective problem with percent energy efficiency ($\in [0, 1]$) being the first objective and the latency ($\in [1, 10]$) in milliseconds being the second objective. Let us also assume that the preference vector ($\boldsymbol{\omega} =$ \{$0.9, 0.1$\}) favors the energy efficiency objective and there are two separate actions ($a_1, a_2$) that yields Q-values of $Q_1 =$\{$0.9, ~1$\}, $Q_2 = $\{$0.1, ~10$\} respectively. The supremum operator chooses action, $a_2$ since $\boldsymbol{\omega}^T Q_2=1.09$ is higher than $\boldsymbol{\omega}^T Q_1=0.91$. Since the scales of these two objectives are in different orders of magnitudes, $\sup_{\substack{a^{\prime}\in A}}(\boldsymbol{\omega}^T\mathbf{Q}s^\prime,a^{\prime},\boldsymbol{\omega}))$ always chooses the action that favors the latency objective. This behavior negatively affects the training since the target Q-values are pulled in the wrong direction.
> > On the contrary, our novel cosine similarity term enables the optimality operator to choose actions that align the preferences with the Q-values and maximize the target value at the same time. The supremum operator now chooses action, $a_1$ since $S_c\cdot\boldsymbol{\omega}^T Q_1=0.75\times0.91 = 0.68$ is higher than $S_c\cdot\boldsymbol{\omega}^T Q_2=0.12\times1.09=0.13$. With this addition, the algorithm disregards the large  $\boldsymbol{\omega}^T\mathbf{Q}(s^\prime,a^{\prime},\boldsymbol{\omega})$ where the preference vector and the Q-values are not aligned since the $S_c(\boldsymbol{\omega},\mathbf{Q}(s^\prime,a^{\prime},\boldsymbol{\omega}))$ takes small values.
> >
> > We have added this detailed explanation of the benefit of using the cosine similarity term in Section 4.2 after Equation 6.
> > The modifications/additions to the original manuscripts are highlighted in blue in the revised manuscript.
> >
> > **C6:** In theorem 4, the authors should define what $\mathbf{Q}^*$ is as it is not clear in the multi-objective setting.
> >
> > **A6:** We thank the reviewer for pointing out the absence of the definition of $\mathbf{Q}^*$. The definition of $\mathbf{Q}^*$ was given in Appendix A.2; however, it was missing in the main manuscript. We have revised the main manuscript and stated explicitly that $\mathbf{Q}^*$ is the unique fixed point, the optimum Q-value for a given preference.

---

> > > ### Author Response · Authors · 2022-12-01
> > > **Response to Reviewer QTgN - Looking forward to hearing your feedback**
> > >
> > > Dear Reviewer QTgN,
> > >
> > > Thanks again for your time and effort, and valued suggestions to improve our work! We have addressed your initial concerns. We are looking forward to hearing your feedback and will be happy to answer/discuss any further questions/concerns you may have.
> > >
> > > Best regards,

---

### Decision · Program_Chairs · 2023-01-20

**Decision:**

Accept: poster

**Justification For Why Not Higher Score:**

None. AC does not have an objection to this being a spotlight.

**Justification For Why Not Lower Score:**

While one reviewer recommended rejecting the paper, their criticisms are either unfounded (e.g., the need to compare to Envelope on the continuous control tasks) as clarified by the author response, or addressed by the authors.

**Metareview: Summary, Strengths And Weaknesses:**

The paper proposes an approach to multi-objective reinforcement learning (MORL) using a single (universal) network that is intended to generalize across the space of preference vectors. The Preference-Driven MORL (PD-MORL) algorithm trains a neural network that takes the preference vector as input (in addition to state) and predict Q-values. A novel contribution of this work is an update to the Bellman update operator that includes the cosine similarity between the preference vector and the Q-value vector. Another is the use of hindsight experience replay (HER) as a means of improving sample efficiency. The paper evaluates PD-MORL on a series of discrete and continuous control domains and shows that it outperforms the Envelop algorithm on discrete tasks while performing comparably to continuous control baselines that must learn separate policies for each preference vector.

The paper received four reviews with overall ratings that range from one reject (3) to two accepts (8). Several of the reviewers agreed on the relevance of MORL to the community and the merits of learning a single network over the space of preference vectors. They also agree that the paper is well written and that the results are readily reproducible due in large part to the inclusion of source code. As two reviewers (Mx6C and sFJy) point out, an additional strength of the paper is the theoretical analysis that proves that the proposed optimality operator is a contraction and that the optimal value function is a fixed point.

The two more negative leaning reviews (QTgN and Mx6C, who gives score of "Marginally Above") are based on concerns that the authors have largely addressed through their response and updates to the paper. Among them, Reviewer QTgN states that the paper should compare against Envelope on the continuous control domains as well as another baseline, however the authors point out that the Envelope implementation only supports discrete environments and that Envelope, which PD-MORL outperforms, has been shown to yield better results than the referenced baseline. Reviewers Mx6C an scaX question the role of the additional cosine similarity term, and Reviewer Mx6C adds that an ablation is necessary to demonstrate its utility. Reviewer Mx6C identifies these as being their major concern with the paper. The authors provide justification for the inclusion of this additional term and point to an existing ablation in the paper.

The authors and the AC made several attempts to engage Reviewers QTgN and Mx6C in discussion, and the AC asked that they update their review to indicate that they had read the author response. Unfortunately. these attempts were unsuccessful. As such the AC is placing more weight on their own reading of the paper and the reviews and discussion provided by Reviewers scaX and sFJy. The AC notes the concerns of these reviewers regarding the clarity of the experimental evaluation related to what they see as only marginal improvements over baselines as well as the need for more insights into settings that exhibit poor performance. The authors are encouraged to revisit the discussion of the experimental results to make sure that the analysis is clearer.

**Note From Pc:**

if the above contains the word "oral" or "spotlight" please see: "oral" presentation means -> notable-top-5% and "spotlight" means -> notable-top-25%. As stated in our emails, we are disassociating presentation type from AC recommendations

**Summary Of Ac-Reviewer Meeting:**

N/A